# Lossy Compression with Gaussian Diffusion

## Abstract

We consider a novel lossy compression approach based on unconditional diffusion generative models, which we call DiffC. Unlike modern compression schemes which rely on transform coding and quantization to restrict the transmitted information, DiffC relies on the efficient communication of pixels corrupted by Gaussian noise. We implement a proof of concept and find that it works surprisingly well despite the lack of an encoder transform, outperforming the state-of-the-art generative compression method HiFiC on ImageNet 64x64. DiffC only uses a single model to encode and denoise corrupted pixels at arbitrary bitrates. The approach further provides support for progressive coding, that is, decoding from partial bit streams. We perform a rate-distortion analysis to gain a deeper understanding of its performance, providing analytical results for multivariate Gaussian data as well as theoretic bounds for general distributions. Furthermore, we prove that a flow-based reconstruction achieves a 3 dB gain over ancestral sampling at high bitrates.

## 1 Introduction

We are interested in the problem of lossy compression with *perfect realism*. As in typical lossy compression applications, our goal is to communicate data using as few bits as possible while simultaneously introducing as little distortion as possible. However, we additionally require that reconstructions $\hat{\mathbf{X}}$ have (approximately) the same marginal distribution as the data, $\hat{\mathbf{X}} \sim \mathbf{X}$. When this constraint is met, reconstructions are indistinguishable from real data or, in other words, appear perfectly realistic. Lossy compression with realism constraints is receiving increasing attention as more powerful generative models bring a solution ever closer within reach. Theoretical arguments (Blau and Michaeli, 2018; 2019; Theis and Agustsson, 2021; Theis and Wagner, 2021) and empirical results (Tschannen et al., 2018; Agustsson et al., 2019; Mentzer et al., 2020) suggest that generative compression approaches have the potential to achieve significantly lower bitrates at similar perceived quality than approaches targeting distortions alone.

The basic idea behind existing generative compression approaches is to replace the decoder with a conditional generative model and to sample reconstructions. Diffusion models (Sohl-Dickstein et al., 2015; Ho et al., 2020)—also known as *score-based generative models* (Song et al., 2021; Dockhorn et al., 2022)—are a class of generative models which have recently received a lot of attention for their ability to generate realistic images (e.g., Dhariwal and Nichol, 2021; Nichol et al., 2021; Ho et al., 2022; Kim and Ye, 2022; Ramesh et al., 2022). While generative compression work has mostly relied on generative adversarial networks (Goodfellow et al., 2014; Tschannen et al., 2018; Agustsson et al., 2019; Mentzer et al., 2020; Gao et al., 2021), Saharia et al. (2021) provided evidence that this approach may also work well with diffusion models by using conditional diffusion models for JPEG artefact removal.

In Section 3, we describe a novel lossy compression approach based on diffusion models. Unlike typical generative compression approaches, our approach relies on an unconditionally trained generative model. Modern lossy compression schemes comprise at least an encoder transform, a decoder transform, and an entropy model (Ballé et al., 2021; Yang et al., 2022) where our approach only uses a single model. Surprisingly, we find that this simple approach can work well despite lacking an encoder transform—instead, we add isotropic Gaussian noise directly to the pixels (Section 5). By using varying degrees of Gaussian noise, the same model can further be used to communicate data at arbitrary bitrates. The approach is naturally progressive, that is, reconstructions can be generated from an incomplete bitstream.

To better understand why the approach works well, we perform a rate-distortion analysis in Section 4. We find that isotropic Gaussian noise is generally not optimal even for the case of Gaussian distributed data and mean-squared error (MSE) distortion. However, we also observe that isotropic noise is close to optimal. We further prove that a reconstruction based on the *probability flow* ODE (Song et al., 2021) cuts the distortion in half at high bit-rates when compared to ancestral sampling from the diffusion model.

We will use capital letters such as $\mathbf{X}$ to denote random variables, lower-case letters such as $\mathbf{x}$ to denote corresponding instances and non-bold letters such as $x_i$ for scalars. We reserve log for the logarithm to base 2 and will use ln for the natural logarithm.

## 2 RELATED WORK

Many previous papers observed connections between variational autoencoders (VAEs; Kingma and Welling, 2014; Rezende et al., 2014) and rate-distortion optimization (e.g., Theis et al., 2017; Ballé et al., 2017; Alemi et al., 2018; Brekelmans et al., 2019; Agustsson and Theis, 2020). More closely related to our approach, Agustsson and Theis (2020) turned a VAE into a practical lossy compression scheme by using dithered quantization to communicate uniform samples. Similarly, our scheme relies on random coding to communicate Gaussian samples and uses diffusion models, which can be viewed as hierarchical VAEs with a fixed encoder.

Ho et al. (2020) considered the rate-distortion performance of an idealized but closely related compression scheme based on diffusion models. In contrast to Ho et al. (2020), we are considering distortion under a perfect realism constraint and provide the first theoretical and empirical results demonstrating that the approach works well. Importantly, random coding is known to provide little benefit and can even hurt performance when only targeting a rate-distortion trade-off (Agustsson and Theis, 2020; Theis and Agustsson, 2021). On the other hand, random codes can perform significantly better than deterministic codes when realism constraints are considered (Theis and Agustsson, 2021). Ho et al. (2020) contemplated the use of *minimal random coding* (MRC; Havasi et al., 2019) to encode Gaussian samples. However, MRC only communicates an approximate sample. In contrast, we consider schemes which communicate an exact sample, allowing us to avoid issues such as error propagation. Finally, we use an upper bound instead of a lower bound as a proxy for the coding cost, which guarantees that our estimated rates are achievable.

While modern lossy compression schemes rely on transform coding, very early work by Roberts (1962) experimented with dithered quantization applied directly to grayscale pixels. Roberts (1962) found that dither was perceptually more pleasing than the banding artefacts caused by quantization. Similarly, we apply Gaussian noise directly to pixels but additionally use a powerful generative model for entropy coding and denoising.

Another line of work in compression explored anisotropic diffusion to denoise and inpaint missing pixels (Galić et al., 2008). This use of diffusion is fundamentally different from ours. Anisotropic diffusion has the effect of smoothing an individual image whereas the diffusion processes considered in this paper are increasing high spatial frequency content of individual images but have a smoothing effect on the distribution over images.

Yan et al. (2021) claimed that under a perfect realism constraint, the best achievable rate is $R(D/2)$, where $R$ is the rate-distortion function (Eq. 7). It was further claimed that optimal performance can be achieved by optimizing an encoder for distortion alone while ignoring the realism constraint and using ancestral sampling at the decoder. Contrary to these claims, we show that our approach can exceed this performance and achieve up to 3 dB better signal-to-noise ratio at the same rate (Figure 2). The discrepancy can be explained by Yan et al. (2021) only considering deterministic codes whereas we allow random codes with access to shared randomness. In random codes, the communicated bits not only depend on the data but are a function of the data and an additional source of randomness shared between the encoder and the decoder (typically implemented by a pseudo-random number generator). Our results are in line with the findings of Theis and Agustsson (2021) who showed on a toy example that shared randomness can lead to significantly better performance in the one-shot setting, and those of Zhang et al. (2021) and Wagner (2022) who studied the rate-distortion-perception function (Blau and Michaeli, 2019) of normal distributions. In this paper, we provide additional results for the multivariate Gaussian case (Section 4.2).

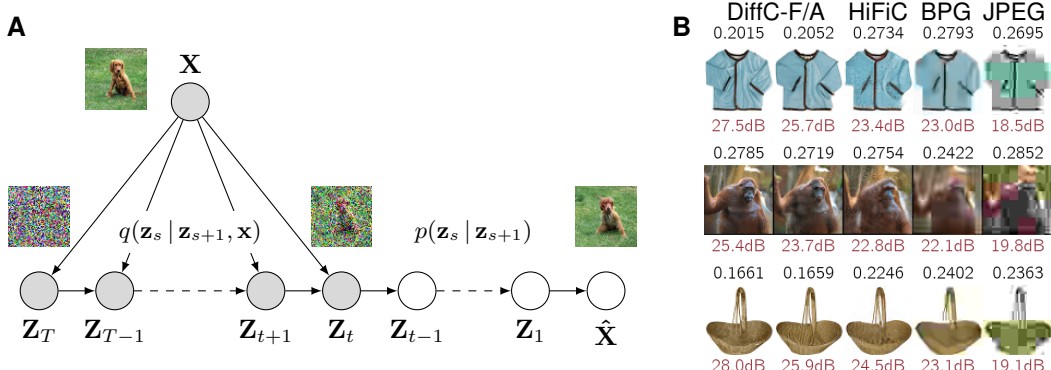

Figure 1: **A**: A visualization of lossy compression with unconditional diffusion models. **B**: Bitrates (bits per pixel; black) and PSNR scores (red) of various approaches including JPEG (4:2:0, headerless) applied to images from the validation set of ImageNet 64x64. For more examples see Appendix I.

An increasing number of neural compression approaches is targeting realism (e.g., Tschannen et al., 2018; Agustsson et al., 2019; Mentzer et al., 2020; Gao et al., 2021; Lytchier et al., 2021; Zeghidour et al., 2022). However, virtually all of these approaches rely on transform coding combined with distortions based on VGG (Simonyan and Zisserman, 2015) and adversarial losses (Goodfellow et al., 2014). In contrast, we use a single unconditionally trained diffusion model (Sohl-Dickstein et al., 2015). Unconditional diffusion models have been used for lossless compression with the help of bits-back coding (Kingma et al., 2021) but bits-back coding by itself is unsuitable for lossy compression. We show that significant bitrate savings can be achieved compared to lossless compression even by allowing imperceptible distortions (Fig. 3).

## 3 LOSSY COMPRESSION WITH DIFFUSION

The basic idea behind our compression approach is to efficiently communicate a corrupted version of the data,

$$\mathbf{Z}_t = \sqrt{1 - \sigma_t^2}\mathbf{X} + \sigma_t\mathbf{U} \quad \text{where} \quad \mathbf{U} \sim \mathcal{N}(0, \mathbf{I}), \tag{1}$$

from the sender to the receiver, and then to use a diffusion generative model to generate a reconstruction. $\mathbf{Z}_t$ can be viewed as the solution to a Gaussian diffusion process given by the stochastic differential equation (SDE)

$$d\mathbf{Z}_t = -\frac{1}{2}\beta_t\mathbf{Z}_t \, dt + \sqrt{\beta_t}d\mathbf{W}_t, \quad \mathbf{Z}_0 = \mathbf{X}, \quad \text{where} \quad \sigma_t^2 = 1 - e^{-\int_0^t \beta_\tau \, d\tau} \tag{2}$$

and $\mathbf{W}_t$ is Brownian motion. Diffusion generative models try to invert this process by learning the conditional distributions $p(\mathbf{z}_s \mid \mathbf{z}_t)$ for $s < t$ (Song et al., 2021). If $s$ and $t$ are sufficiently close, then this conditional distribution is approximately Gaussian. We refer to Sohl-Dickstein et al. (2015), Ho et al. (2020), and Song et al. (2021) for further background on diffusion models.

Noise has a negative effect on the performance of typical compression schemes (Al-Shaykh and Mersereau, 1998). However, Bennett and Shor (2002) proved that it is possible to communicate an instance of $\mathbf{Z}_t$ using not much more than $I[\mathbf{X}, \mathbf{Z}_t]$ bits. Note that this mutual information decreases as the level of noise increases. Li and El Gamal (2018) described a more concrete random coding approach for communicating an exact sample of $\mathbf{Z}_t$ (Appendix A). An upper bound was provided for its coding cost, namely

$$I[\mathbf{X}, \mathbf{Z}_t] + \log(I[\mathbf{X}, \mathbf{Z}_t] + 1) + 5 \tag{3}$$

bits. Notice that the second and third term become negligible when the mutual information is sufficiently large. If the sender and receiver do not have access to the true marginal of $\mathbf{Z}_t$ but instead

assume the marginal distribution to be $p_t$, the upper bound on the coding cost becomes (Theis and Yosri, 2022)

$$C_t + \log(C_t + 1) + 5 \quad \text{where} \quad C_t = \mathbb{E}_{\mathbf{X}}[D_{\text{KL}}[q(\mathbf{z}_t \mid \mathbf{X}) \parallel p_t(\mathbf{z}_t)]] \tag{4}$$

and $q$ is the distribution of $\mathbf{Z}_t$ given $\mathbf{X}$, which in our case is Gaussian. In practice, the coding cost can be significantly closer to $C_t$ than the upper bound (Theis and Yosri, 2022; Flamich et al., 2022).

We refer to Theis and Yosri (2022) for an introduction to the problem of efficient sample communication—also known as *reverse channel coding*—as well as a discussion of practical implementations of the approach of Li and El Gamal (2018). To follow the results of this paper, the reader only needs to know that an exact sample of a distribution $q$ can be communicated with a number of bits which is at most the bound given in Eq. 4, and that this is possible even when $q$ is continuous. The bound above is analogous to the well-known result that the cost of entropy coding can be bounded in terms of $H + 1$, where $H$ is a cross-entropy (e.g., Cover and Thomas, 2006). However, to provide some intuition for reverse channel coding, we briefly describe the high-level idea. Candidates $\mathbf{Z}_t^1, \mathbf{Z}_t^2, \mathbf{Z}_t^3, \ldots$ are generated by drawing samples from $p_t$. The encoder then selects one of the candidates with index $N^*$ in a manner similar to rejection sampling such that $\mathbf{Z}_t^{N^*} \sim q$. Since the candidates are independent of the data, they can be generated by both the sender and receiver (for example, using a pseudo-random number generator with the same random seed) and only the selected candidate's index $N^*$ needs to be communicated. The entropy of $N^*$ is bounded by Eq. 4. Further details and pseudocode are provided in Appendix A

Unfortunately, Gaussian diffusion models do not provide us with tractable marginal distributions $p_t$. Instead, they give us access to conditional distributions $p(\mathbf{z}_s \mid \mathbf{z}_{s+1})$ and assume $p_T$ is isotropic Gaussian. This suggests a scheme where we first transmit an instance of $\mathbf{Z}_T$ and then successively refine the information received by the sender by transmitting an instance of $\mathbf{Z}_s$ given $\mathbf{Z}_{s+1}$ until $\mathbf{Z}_t$ is reached. This approach incurs an overhead for the coding cost of each conditional sample (which we consider in Fig. 10 of Appendix I). Alternatively, we can communicate a Gaussian sample from the joint distribution $q(\mathbf{z}_{T:t} \mid \mathbf{X})$ directly while assuming a marginal distribution $p(\mathbf{z}_{T:t})$. This achieves a coding cost upper bounded by Eq. 4 where

$$C_t = \mathbb{E}\left[D_{\text{KL}}[q(\mathbf{z}_T \mid \mathbf{X}) \parallel p_T(\mathbf{z}_T)]\right] + \sum_{s=1}^{T-1} \mathbb{E}\left[D_{\text{KL}}[q(\mathbf{z}_s \mid \mathbf{Z}_{s+1}, \mathbf{X}) \parallel p(\mathbf{z}_s \mid \mathbf{Z}_{s+1})]\right]. \tag{5}$$

Reverse channel coding still poses several unsolved challenges in practice. In particular, the scheme proposed by Li and El Gamal (2018) is computationally expensive though progress on more efficient schemes is being made (Agustsson and Theis, 2020; Theis and Yosri, 2022; Flamich et al., 2022). In the following we will mostly ignore issues of computational complexity and instead focus on the question of whether the approach described above is worth considering at all. After all, it is not immediately clear that adding isotropic Gaussian noise directly to the data would limit information in a useful way.

We will consider two alternatives for reconstructing data given $\mathbf{Z}_t$. First, we will consider ancestral sampling, $\hat{\mathbf{X}} \sim p(\mathbf{x} \mid \mathbf{Z}_t)$, which corresponds to simulating the SDE in Eq. 2 in reverse (Song et al., 2021). Second, we will consider a deterministic reconstruction which instead tries to reverse the ODE

$$d\mathbf{z}_t = \left(-\frac{1}{2}\beta_t \mathbf{z}_t - \frac{1}{2}\beta_t \nabla \ln p_t(\mathbf{z}_t)\right) dt. \tag{6}$$

Maoutsa et al. (2020) and Song et al. (2021) showed that this "probability flow" ODE produces the same trajectory of marginal distributions $p_t$ as the Gaussian diffusion process in Eq. 2 and that it can be simulated using the same model of $\nabla \ln p_t(\mathbf{z}_t)$. We will refer to these alternatives as DiffC-A when ancestral sampling is used and DiffC-F when the flow-based reconstruction is used.

## 4 A RATE-DISTORTION ANALYSIS

In this section we try to understand the performance of DiffC from a rate-distortion perspective. This will be achieved by considering the Gaussian case where optimal rate-distortion trade-offs can be computed analytically and by providing bounds on the performance in the general case. Throughout this paper, we measure distortion in terms of squared error. For our theoretical analysis we will further assume that the diffusion model has learned the data distribution perfectly.

The (information) rate-distortion function is given by

$$R(D) = \inf_{\hat{\mathbf{X}}} I[\mathbf{X}, \hat{\mathbf{X}}] \quad \text{subject to} \quad \mathbb{E}[\|\mathbf{X} - \hat{\mathbf{X}}\|^2] \leq D. \tag{7}$$

It measures the smallest achievable bitrate for a given level of distortion and decreases as $D$ increases[1].

The rate as defined above does not make any assumptions on the marginal distribution of the reconstructions. However, here we demand perfect realism, that is, $\hat{\mathbf{X}} \sim \mathbf{X}$. To achieve this constraint, a deterministic encoder requires a higher bitrate of $R(D/2)$ (Blau and Michaeli, 2019; Theis and Agustsson, 2021). As we will see below, lower bitrates can be achieved using random codes as in our diffusion approach. Nevertheless, $R(D/2)$ serves as an interesting benchmark as most existing codecs use deterministic codes, that is, the bits received by the decoder are solely determined by the data.

For an $M$-dimensional Gaussian data source whose covariance has eigenvalues $\lambda_i$, the rate-distortion function is known to be (Cover and Thomas, 2006)

$$R^*(D) = \tfrac{1}{2} \sum_i \log(\lambda_i/D_i) \quad \text{where} \quad D_i = \min(\lambda_i, \theta) \tag{8}$$

for some threshold $\theta$ chosen such that $D = \sum_i D_i$. For sufficiently small distortion $D$ and assuming positive eigenvalues, we have constant $D_i = \theta = D/M$.

### 4.1 STANDARD NORMAL DISTRIBUTION

As a simple first example, consider a standard normal distribution $X \sim \mathcal{N}(0, 1)$. Using ancestral sampling, the reconstruction becomes

$$\hat{X} = \sqrt{1 - \sigma^2} Z + \sigma V \quad \text{where} \quad Z = \sqrt{1 - \sigma^2} X + \sigma U, \tag{9}$$

$U, V \sim \mathcal{N}(0, 1)$ and we have dropped the dependence on $t$ to reduce clutter. The distortion and rate in this case are easily calculated to be

$$D = \mathbb{E}[(X - \hat{X})^2] = 2\sigma^2, \qquad I[X, Z] = -\log \sigma = \frac{1}{2} \log \frac{2}{D} = R^*(D/2). \tag{10}$$

This matches the performance of an optimal deterministic code. However, $Z$ already has the desired standard normal distribution and adding further noise to it did nothing to increase the realism or reduce the distortion of the reconstruction. The flow-based reconstruction instead yields $dZ_t = 0$ and $\hat{X} = Z$ (by inserting the standard normal for $p_t$ in Eq. 6), resulting in the smaller distortion

$$D = \mathbb{E}[(X - \hat{X})^2] = \mathbb{E}[(X - Z)^2] = 2 - 2\sqrt{1 - \sigma^2}. \tag{11}$$

### 4.2 MULTIVARIATE GAUSSIAN

Next, let us consider $\mathbf{X} \sim \mathcal{N}(0, \boldsymbol{\Sigma})$ and $\mathbf{Z} = \sqrt{1 - \sigma^2} \mathbf{X} + \sigma \mathbf{U}$ where $\mathbf{U} \sim \mathcal{N}(0, \mathbf{I})$. Assume $\lambda_i$ are the eigenvalues of $\boldsymbol{\Sigma}$. Since both the squared reconstruction error and the mutual information between $\hat{\mathbf{X}}$ and $\mathbf{Z}$ are invariant under rotations of $\mathbf{X}$, we can assume the covariance to be diagonal. Otherwise we just rotate $\mathbf{X}$ to diagonalize the covariance matrix without affecting the results of our analysis. If $\hat{\mathbf{X}} \sim P(\mathbf{X} \,|\, \mathbf{Z})$, we get the distortion and rate (Appendix C)

$$D = \mathbb{E}[\|\mathbf{X} - \hat{\mathbf{X}}\|^2] = 2 \sum_i \tilde{D}_i, \qquad I[\mathbf{X}, \mathbf{Z}] = \tfrac{1}{2} \sum_i \log(\lambda_i/\tilde{D}_i) \geq R^*(D/2). \tag{12}$$

where $\tilde{D}_i = \lambda_i \sigma^2 / (\sigma^2 + \lambda_i - \lambda_i \sigma^2)$. That is, the performance is generally worse than the performance achieved by the best deterministic encoder. We can modify the diffusion process to improve the rate-distortion performance of ancestral sampling. Namely, let $V_i \sim \mathcal{N}(0, 1)$,

$$Z_i = \sqrt{1 - \gamma_i^2} X_i + \gamma_i \sqrt{\lambda_i} U_i, \qquad \hat{X}_i = \sqrt{1 - \gamma_i^2} Z_i + \gamma_i \sqrt{\lambda_i} V_i, \tag{13}$$

where $\gamma_i^2 = \min(1, \theta/\lambda_i)$ for some $\theta$. This amounts to using a different noise schedule along different principal directions instead of adding the same amount of noise in all directions. For natural images,

---

[1]The bitrate given by an information rate-distortion may only be achievable asymptotically by encoding many data points jointly. To keep our discussion focused, we ignore any potential overhead incurred by *one-shot coding* and use mutual information as a proxy for the rate achieved in practice.

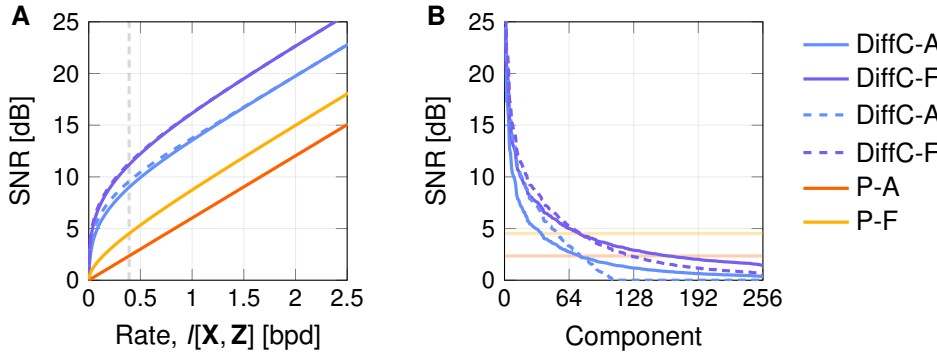

Figure 2: **A**: Rate-distortion curves for a Gaussian source fitted to 16x16 image patches extracted from ImageNet 64x64. Isotropic noise performs nearly as well as the optimal noise (dashed). As an additional point of comparison, we include pink noise (P) matching the covariance of the data distribution. The curve of DiffC-A* corresponds to $R^*(D/2)$. A flow-based reconstruction yields up to 3 dB better signal-to-noise ratio (SNR). **B**: SNR broken down by principal component. The level of noise here is fixed to yield a rate of approximately 0.391 bits per dimension for each type of noise. Note that the SNR of DiffC-A* is zero for over half of the components.

the modified schedule destroys information in high-frequency components more quickly (Fig. 2B) and for Gaussian data sources again matches the performance of the best deterministic code,

$$D = 2\sum_i \lambda_i \gamma_i^2 = 2\sum_i D_i, \quad I[\mathbf{X}, \mathbf{Z}] = -\sum_i \log \gamma_i = \frac{1}{2}\sum_i \log(\lambda_i/D_i) = R^*(D/2) \quad (14)$$

where $D_i = \lambda_i \gamma_i^2 = \min(\lambda_i, \theta)$. Still better performance can be achieved via flow-based reconstruction. Here, isotropic noise is again suboptimal and the optimal noise for a flow-based reconstruction is given by (Appendix D)

$$Z_i = \alpha_i X_i + \sqrt{1-\alpha_i^2}\sqrt{\lambda_i}U_i, \quad \text{where} \quad \alpha_i = \left(\sqrt{\lambda_i^2 + \theta^2} - \theta\right)/\lambda_i \quad (15)$$

for some $\theta \geq 0$. $\mathbf{Z}$ already has the desired distribution and we can set $\hat{\mathbf{X}} = \mathbf{Z}$.

We will refer to the two approaches using optimized noise as DiffC-A* and DiffC-F*, respectively, though strictly speaking these types of noise may no longer correspond to diffusion processes. Figure 2A shows the rate-distortion performance of the various noise schedules and reconstructions on the example of a 256-dimensional Gaussian fitted to 16x16 grayscale image patches extracted from 64x64 downsampled ImageNet images (van den Oord et al., 2016). Here, SNR $= 10\log_{10}(2 \cdot \mathbb{E}[\|\mathbf{X}\|^2]) - 10\log_{10}(\mathbb{E}[\|\mathbf{X} - \hat{\mathbf{X}}\|^2])$.

### 4.3 GENERAL DATA DISTRIBUTIONS

Considering more general source distributions, our first result bounds the rate of DiffC-A*.

**Theorem 1.** *Let* $\mathbf{X} : \Omega \to \mathbb{R}^M$ *be a random variable with finite differential entropy, zero mean and covariance* $\text{diag}(\lambda_1, \ldots, \lambda_M)$*. Let* $\mathbf{U} \sim \mathcal{N}(0, \mathbf{I})$ *and define*

$$Z_i = \sqrt{1-\gamma_i^2}X_i + \gamma_i\sqrt{\lambda_i}U_i, \qquad \hat{\mathbf{X}} \sim P(\mathbf{X} \mid \mathbf{Z}). \quad (16)$$

*where* $\gamma_i^2 = \min(1, \theta/\lambda_i)$ *for some* $\theta$*. Further, let* $\mathbf{X}^*$ *be a Gaussian random variable with the same first and second-order moments as* $\mathbf{X}$ *and let* $\mathbf{Z}^*$ *be defined analogously to* $\mathbf{Z}$ *but in terms of* $\mathbf{X}^*$*. Then if* $R$ *is the rate-distortion function of* $\mathbf{X}$ *and* $R^*$ *is the rate-distortion function of* $\mathbf{X}^*$*,*

$$I[\mathbf{X}, \mathbf{Z}] \leq R^*(D/2) - D_{\text{KL}}[P_{\mathbf{Z}} \| P_{\mathbf{Z}^*}] \leq R(D/2) + D_{\text{KL}}[P_{\mathbf{X}} \| P_{\mathbf{X}^*}] - D_{\text{KL}}[P_{\mathbf{Z}} \| P_{\mathbf{Z}^*}] \quad (17)$$

*where* $D = \mathbb{E}[\|\mathbf{X} - \hat{\mathbf{X}}\|^2]$*.*

*Proof.* See Appendix E. □

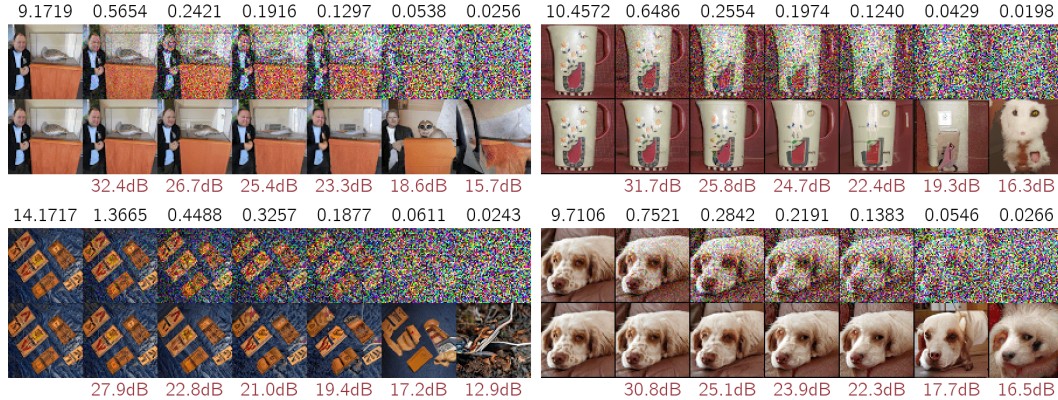

Figure 3: Top images visualize messages communicated at the estimated bitrate (bits per pixel) shown in black. The bottom row shows reconstructions produced by DiffC-F and corresponding PSNR values are shown in red.

In line with expectations, this result implies that when $\mathbf{X}$ is approximately Gaussian, the rate of DiffC-A$^*$ is not far from the rate of the best deterministic encoder, $R(D/2)$. It further implies that the rate is close to $R(D/2)$ in the high bitrate regime if the differential entropy of $\mathbf{X}$ is finite. This can be seen by noting that the second KL divergence will approach the first KL divergence as the rate increases, since $P_{\mathbf{Z}^*} = P_{\mathbf{X}^*}$ and the distribution of $\mathbf{Z}$ will be increasingly similar to $\mathbf{X}$.

Our next result compares the error of DiffC-F with DiffC-A's at the same bitrate. For simplicity, we assume that $\mathbf{X}$ has a smooth density and further consider the following measure of smoothness,

$$G = \mathbb{E}\left[\|\nabla \ln p(\mathbf{X})\|^2\right]. \tag{18}$$

Among distributions with a continuously differentiable density and unit variance, the standard normal distribution minimizes $G$ and achieves $G = 1$. For comparison, the Laplace distribution has $G = 2$. (Alternatively, imagine a sequence of smooth approximations converging to the Laplace density.) For discrete data such as RGB images, we may instead consider the distribution of pixels with an imperceptible amount of Gaussian noise added to it (see also Fig. 5 in Appendix F).

**Theorem 2.** *Let $\mathbf{X} : \Omega \to \mathbb{R}^M$ have a smooth density $p$ with finite $G$ (Eq. 18). Let $\mathbf{Z}_t$ be defined as in Eq. 1, $\hat{\mathbf{X}}_A \sim P(\mathbf{X} \mid \mathbf{Z}_t)$ and let $\hat{\mathbf{X}}_F = \hat{\mathbf{Z}}_0$ be the solution to Eq. 6 with $\mathbf{Z}_t$ as initial condition. Then*

$$\lim_{\sigma_t \to 0} \frac{\mathbb{E}[\|\hat{\mathbf{X}}_F - \mathbf{X}\|^2]}{\mathbb{E}[\|\hat{\mathbf{X}}_A - \mathbf{X}\|^2]} = \frac{1}{2} \tag{19}$$

*Proof.* See Appendix F. $\square$

This result implies that in the limit of high bitrates, the error of a flow-based reconstruction is only half that of the the reconstruction obtained with ancestral sampling from a perfect model. This is consistent with Fig. 2, where we can observe an advantage of roughly 3 dB of DiffC-F over DiffC-A. Finally, we provide conditions under which a flow-based reconstruction is provably the best reconstruction from input corrupted by Gaussian noise.

**Theorem 3.** *Let $\mathbf{X} = \mathbf{QS}$ where $\mathbf{Q}$ is an orthogonal matrix and $\mathbf{S} : \Omega \to \mathbb{R}^M$ is a random vector with smooth density and $S_i \perp\!\!\!\perp S_j$ for all $i \neq j$. Define $\mathbf{Z}_t$ as in Eq. 1. If $\hat{\mathbf{X}}_F = \hat{\mathbf{Z}}_0$ is the solution to the ODE in Eq. 6 given $\mathbf{Z}_t$ as initial condition, then*

$$\mathbb{E}[\|\hat{\mathbf{X}}_F - \mathbf{X}\|^2] \leq \mathbb{E}[\|\hat{\mathbf{X}}' - \mathbf{X}\|^2] \tag{20}$$

*for any $\hat{\mathbf{X}}'$ with $\hat{\mathbf{X}}' \perp\!\!\!\perp \mathbf{X} \mid \mathbf{Z}_t$ which achieves perfect realism, $\hat{\mathbf{X}}' \sim \mathbf{X}$.*

*Proof.* See Appendix G. $\square$

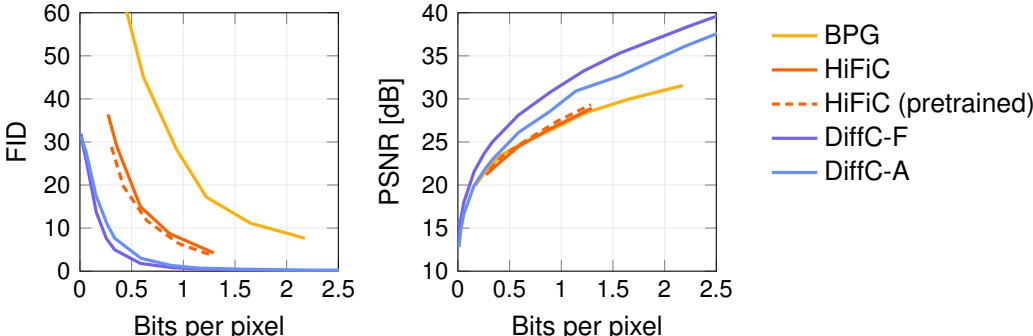

Figure 4: A comparison of DiffC with BPG and the GAN-based neural compression method HiFiC in terms of FID and PSNR on ImageNet 64x64.

## 5 EXPERIMENTS

As a proof of concept, we implemented DiffC based on VDM[2] (Kingma et al., 2021). VDM is a diffusion model which was optimized for log-likelihood (i.e., lossless compression) but not for perceptual quality. This suggests VDM should work well in the high bitrate regime but not necessarily at lower bitrates. Nevertheless, we find that we achieve surprisingly good performance across a wide range of bitrates. We used exactly the same network architecture and training setup as Kingma et al. (2021) except with a smaller batch size of 64 images and training our model for only 1.34M updates (instead of 2M updates with a batch size of 512) due to resource considerations. We used 1000 diffusion steps.

### 5.1 DATASET, METRICS, AND BASELINES

We used the downsampled version of the ImageNet dataset (Deng et al., 2009) (64x64 pixels) first used by van den Oord et al. (2016). The test set of ImageNet is known to contain many duplicates and to overlap with the training set (Kolesnikov et al., 2019). For a more meaningful evaluation (especially when comparing to non-neural baselines), we removed 4952 duplicates from the validation set as well as 744 images also occuring in the training set (based on SHA-256 hashes of the images). On this subset, we measured a negative ELBO of 3.48 bits per dimension for our model.

We report FID (Heusel et al., 2017) and PSNR scores to quantify the performance of the different approaches. As is common in the compression literature, in this section we calculate a PSNR score for each image before averaging. For easier comparison with our theoretical results, we also offer PSNR scores calculated from the average MSE (Appendix I) although the numbers do not change markedly. When comparing bitrates between models, we used estimates of the upper bound given by Eq. 4 for DiffC.

We compare against BPG (Bellard, 2018), a strong non-neural image codec based on the HEVC video codec which is known for achieving good rate-distortion results. We also compare against HiFiC (Mentzer et al., 2020), which is the state-of-the-art generative image compression model in terms of visual quality on high-resolution images. The approach is optimized for a combination of LPIPS (Zhang et al., 2018), MSE, and an adversarial loss (Goodfellow et al., 2014). The architecture of HiFiC is optimized for larger images and uses significant downscaling. We found that adapting the architecture of HiFiC slightly by making the last/first layer of the encoder/decoder have stride 1 instead of stride 2 improves FID on ImageNet 64x64 compared to the publicly available model. In addition to training the model from scratch, we also tried initializing the non-adapted filters from the public model and found that this improved results slightly. We trained 5 HiFiC models targeting 5 different bitrates.

---

[2]https://github.com/google-research/vdm

## 5.2 Results

We find that DiffC-F gives perceptually pleasing results even at extremely low bitrates of around 0.2 bits per pixel (Fig. 3). Reconstructions are also still perceptually pleasing when the PSNR is relatively low at around 22 dB (e.g., compare to BPG in Fig. 1B). We further find that at very low bitrates, HiFiC produces artefacts typical for GANs while we did not observe similar artefacts with DiffC. Similar conclusions can be drawn from our quantitative comparison, with DiffC-F significantly outperforming HiFiC in terms of FID. FID scores of DiffC-A were only slightly worse (Fig. 4A).

At high bitrates, DiffC-F achieves a PSNR roughly 2.4 dB higher than DiffC-A. This is line with our theoretical predictions (3 dB) considering that the diffusion model only approximates the true distribution. PSNR values of DiffC-F and DiffC-A both exceed those of HiFiC and BPG, suggesting that Gaussian diffusion works well in a rate-distortion sense even for highly non-Gaussian distributions (Fig. 4B). Additional results are provided in Appendix I, including results for progressive coding and HiFiC trained for MSE only.

## 6 Discussion

We presented and analyzed a new lossy compression approach based on diffusion models. This approach has the potential to greatly simplify lossy compression with realism constraints. Where typical generative approaches use an encoder, a decoder, an entropy model, an adversarial model and another model as part of a perceptual distortion loss, and train multiple sets of models targeting different bitrates, DiffC only uses a single unconditionally trained diffusion model. The fact that adding Gaussian noise to pixels achieves great rate-distortion performance raises interesting questions about the role of the encoder transform in lossy compression. Nevertheless, we expect further improvements are possible in terms of perceptual quality by applying DiffC in a latent space.

Applying DiffC in a lower-dimensional transform space would also help to reduce its computational cost (Vahdat et al., 2021; Rombach et al., 2021; Gu et al., 2021; Pandey et al., 2022). The high computational cost of DiffC makes it impractical in its current form. Generating a single image with VDM requires many diffusion steps, each involving the application of a deep neural network. However, speeding up diffusion models is a highly active area of research (e.g., Watson et al., 2021; Vahdat et al., 2021; Jolicoeur-Martineau et al., 2021; Kong and Ping, 2021; Salimans and Ho, 2022; Zhang and Chen, 2022). For example, Salimans and Ho (2022) were able to reduce the number of diffusion steps from 1000 to around 4 at comparable sample quality. The computational cost of communicating a sample using the approach of Li and El Gamal (2018) grows exponentially with the coding cost. However, reverse channel coding is another active area of research (e.g., Havasi et al., 2019; Agustsson and Theis, 2020; Flamich et al., 2020) and much faster methods already exist for low-dimensional Gaussian distributions (Theis and Yosri, 2022; Flamich et al., 2022). Our work offers strong motivation for further research into more efficient reverse channel coding schemes.

As mentioned in Section 3, reverse channel coding may be applied after each diffusion step to send a sample of $q(\mathbf{z}_t \mid \mathbf{Z}_{t+1}, \mathbf{X})$, or alternatively to the joint distribution $q(\mathbf{z}_{T:t} \mid \mathbf{X})$. The former approach has the advantage of lower computational cost due to the exponential growth with the coding cost. Furthermore, the model's score function only needs to be evaluated once per diffusion step to compute a conditional mean while the latter approach requires many more evaluations (one for each candidate considered by the reverse channel coding scheme). Fig. 10 shows that this approach—which is already much more practical—still significantly outperforms HiFiC. Another interesting avenue to consider is replacing Gaussian $q(\mathbf{z}_t \mid \mathbf{Z}_{t+1}, \mathbf{X})$ with a uniform distribution, which can be simulated very efficiently (e.g., Zamir and Feder, 1996; Agustsson and Theis, 2020).

We provided an initial theoretical analysis of DiffC. In particular, we analyzed the Gaussian case and proved that DiffC-A* performs well when either the data distribution is close to Gaussian or when the bitrate is high. In particular, the rate of DiffC-A* approaches $R(D/2)$ at high birates. We further proved that DiffC-F can achieve 3 dB better SNR at high bitrates compared to DiffC-A. Taken together, these results suggest that $R(D)$ may be achievable at high bitrates where current approaches based on nonlinear transform coding can only achieve $R(D/2)$. However, many theoretical questions have been left for future research. For instance, how does the performance of DiffC-A differ from DiffC-A*? And can we extend Theorem 3 to prove optimality of a flow-based reconstruction from noisy data for a broader class of distributions?

## REPRODUCIBILITY STATEMENT

Our appendix provides extensive proofs of the claims made in our paper. Every effort has been made to make explicit the assumptions in our claims. The models used in our empirical results (VDM, HiFiC) are based on open source code. We further provide code to reproduce the results of Fig. 2.

## ETHICS STATEMENT

Lossy compression achieving perfect realism produces reconstructions which are indistinguishable from real data and which hide a loss of information. This property is desirable in applications such as video streaming for entertainment purposes but can be problematic in applications such as surveillance, medical imaging, or document compression where the image content influences critical decisions. The availability of better generative compression methods increases the risk of misuse in these areas.

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

## A    REVERSE CHANNEL CODING

---

**Algorithm 1** Encoding (Li and El Gamal, 2018; Theis and Yosri, 2022)

---

**Require:** $p, q, w_{\min}$
1: $t, n, s^* \leftarrow 0, 1, \infty$

2: **repeat**
3:     $z \leftarrow \texttt{simulate}(n, p)$
4:     $t \leftarrow t + \texttt{exponential}(1)$
5:     $s \leftarrow t \cdot p(z)/q(z)$

6:     **if** $s < s^*$ **then**
7:         $s^*, n^* \leftarrow s, n$
8:     **end if**

9:     $n \leftarrow n + 1$
10: **until** $s^* \leq t \cdot w_{\min}$

11: **return** $n^*$

---

---

**Algorithm 2** Decoding

---

**Require:** $n^*, \text{p}$
 1: **return** `simulate`$(n^*, \text{p})$

---

For completeness, we here reproduce pseudocode by Theis and Yosri (2022) of the sampling scheme first considered by Maddison (2016) and later by Li and El Gamal (2018) for the purpose of reverse channel coding. Similar to rejection sampling, the encoding process accepts a candidate generating distribution $p$, a target distribution $q$, and a bound on the density ratio

$$w_{\min} \leq \inf_{\mathbf{z}} \frac{p(\mathbf{z})}{q(\mathbf{z})}. \tag{21}$$

The encoding process returns a index $N^*$ (random due to the exponential noise) such that $Z_{N^*}$ follows the distribution $q$ (Algorithm 1). Importantly, the algorithm produces an *exact* sample in a finite number of steps (Maddison, 2016). Furthermore, the coding cost of $N^*$ is bounded by (Li and El Gamal, 2018)

$$H[N^*] + 1 < I[\mathbf{X}, \mathbf{Z}] + \log(I[\mathbf{X}, \mathbf{Z}] + 1) + 5 \tag{22}$$

and this bound can be achieved by entropy encoding $N^*$ with a Zipf distribution $p_\lambda(n) \propto n^{-\lambda}$ which has a single parameter

$$\lambda = 1 + \frac{1}{I[\mathbf{X}, \mathbf{Z}] + e^{-1} \log e + 1}. \tag{23}$$

In practice, the coding cost for Gaussians may be significantly lower than this bound.

In the encoding process, the function `simulate`$(n, \text{p})$ returns the $n$th candidate $\mathbf{Z}_n \sim p$ (in practice, this would be achieved with a pseudo-random number generator though we could also imagine a large list of previously generated and shared samples). The function `exponential`$(1)$ produces a random sample from an exponential distribution with rate 1.

Unlike encoding, decoding is fast as it only amounts to selecting the right candidate once $N^*$ has been received (Algorithm 2).

## B   NORMAL DISTRIBUTION WITH NON-UNIT VARIANCE

The case of a 1-dimensional Gaussian with varying variance is essentially the same as for a standard normal. In both cases, a Gaussian source is communicated through a Gaussian channel. Let $X \sim \mathcal{N}(0, \lambda)$ and

$$Z = \sqrt{1 - \sigma^2} X + \sigma U \tag{24}$$

where as before $U \sim \mathcal{N}(0, 1)$. Let $\hat{X} \sim P(X \mid Z)$. We define

$$\tilde{\sigma}^2 = \frac{\sigma^2}{\sigma^2 + \lambda - \lambda \sigma^2}. \tag{25}$$

Then

$$I[X, Z] = h[Z] - h[Z \mid X] = \frac{1}{2} \log \left( \lambda - \sigma^2 \lambda + \sigma^2 \right) - \frac{1}{2} \log \sigma^2 = -\log \tilde{\sigma} \tag{26}$$

and

$$D = \mathbb{E}[(X - \hat{X})^2] = \lambda \mathbb{E}[(\lambda^{-\frac{1}{2}} X - \lambda^{-\frac{1}{2}} \hat{X})^2] = 2\lambda \tilde{\sigma}^2 \tag{27}$$

due to Eq. 10 which tells us the squared error of the standard normal $\lambda^{-\frac{1}{2}} X$ as a function of the information contained in $Z$. Taken together, we again have

$$I[X, Z] = \frac{1}{2} \log \frac{\lambda}{D/2} = R^*(D/2). \tag{28}$$

## C   Multivariate Gaussian

Let $\mathbf{X} \sim \mathcal{N}(0, \boldsymbol{\Sigma})$ and let

$$\mathbf{Z} = \sqrt{1 - \sigma^2}\mathbf{X} + \sigma\mathbf{U} \tag{29}$$

where $\mathbf{U} \sim \mathcal{N}(0, \mathbf{I})$. Note that both the mutual information and the squared error are invariant under rotations of $\mathbf{X}$. We can therefore assume that the covariance is diagonal,

$$\boldsymbol{\Sigma} = \mathrm{diag}(\lambda_1, \dots, \lambda_M). \tag{30}$$

Defining

$$\tilde{\sigma}_i^2 = \frac{\sigma^2}{\sigma^2 + \lambda_i - \lambda_i\sigma^2}. \tag{31}$$

as in Appendix B, we have

$$D = \mathbb{E}[\|\mathbf{X} - \hat{\mathbf{X}}\|^2] = \sum_i \mathbb{E}[(X_i - \hat{X}_i)^2] = 2\sum_i \lambda_i\tilde{\sigma}_i^2 \tag{32}$$

and

$$I[\mathbf{X}, \mathbf{Z}] = \sum_i I[X_i, Z_i] = -\frac{1}{2}\sum_i \log \tilde{\sigma}_i^2 \tag{33}$$

Let $\tilde{D}_i = \lambda_i\tilde{\sigma}_i^2$, then we can write

$$D = 2\sum_i \tilde{D}_i, \qquad\qquad I[\mathbf{X}, \mathbf{Z}] = \frac{1}{2}\sum_i \log \frac{\lambda_i}{\tilde{D}_i}. \tag{34}$$

For fixed distortion $D$, the rate in Eq. 34 as a function of $\tilde{D}_i$ is known to be minimized by the so-called reverse water-filling solution given in Eq. 8 (Shannon, 1949; Cover and Thomas, 2006), that is, $D_i = \min(\lambda_i, \theta)$ where $\theta$ must be chosen so that $D = 2\sum_i D_i$. Hence, $\tilde{D}_i$ as defined above is generally suboptimal and we must have

$$I[\mathbf{X}, \mathbf{Z}] \geq \frac{1}{2}\sum_i \log \frac{\lambda_i}{D_i} = R^*(D/2). \tag{35}$$

## D   Optimal noise schedule for flow-based reconstruction

**Lemma 1.** *Let $\mathbf{X} \sim \mathcal{N}(0, \boldsymbol{\Sigma})$ with diagonal covariance matrix*

$$\boldsymbol{\Sigma} = diag(\lambda_1, \dots, \lambda_M) \tag{36}$$

*and $\lambda_i > 0$. Further let*

$$Z_i = \alpha_i X_i + \sqrt{1 - \alpha_i^2}\sqrt{\lambda_i}U_i, \qquad\qquad \hat{\mathbf{X}} = \mathbf{Z}, \tag{37}$$

*where $\mathbf{U} \sim \mathcal{N}(0, \mathbf{I})$ and*

$$\alpha_i = \left(\sqrt{\lambda_i^2 + \theta^2} - \theta\right)/\lambda_i. \tag{38}$$

*Then $\hat{\mathbf{X}}$ achieves the minimal rate at distortion level $D = \mathbb{E}[\|\mathbf{X} - \hat{\mathbf{X}}\|^2]$ among all reconstructions satisfying the realism constraint $\hat{\mathbf{X}} \sim \mathbf{X}$.*

*Proof.* The lowest rate achievable by a code (with access to a source of shared randomness) is (Theis and Wagner, 2021)

$$\inf_{\tilde{\mathbf{X}}} I[\mathbf{X}, \tilde{\mathbf{X}}] \quad \text{subject to} \quad \mathbb{E}[\|\mathbf{X} - \tilde{\mathbf{X}}\|] \leq D \quad \text{and} \quad \mathbf{X} \sim \tilde{\mathbf{X}}. \tag{39}$$

We can rewrite the rate as

$$\inf_{\tilde{\mathbf{X}}} I[\mathbf{X}, \tilde{\mathbf{X}}] = \inf_{\tilde{\mathbf{X}}} h[\mathbf{X}] + h[\tilde{\mathbf{X}}] - h[\mathbf{X}, \tilde{\mathbf{X}}] = 2h[\mathbf{X}] - \sup_{\tilde{\mathbf{X}}} h[\mathbf{X}, \tilde{\mathbf{X}}]. \tag{40}$$

That is, we need to maximize the differential entropy of $(\mathbf{X}, \tilde{\mathbf{X}})$ subject to constraints. For the distortion constraint, we have

$$\mathbb{E}[\|\mathbf{X} - \tilde{\mathbf{X}}\|^2] = \mathbb{E}[\mathbf{X}^\top \mathbf{X}] + \mathbb{E}[\tilde{\mathbf{X}}^\top \tilde{\mathbf{X}}] - 2\mathbb{E}[\mathbf{X}^\top \tilde{\mathbf{X}}] = 2\sum_i \lambda_i - 2\sum_i \mathbb{E}[X_i \tilde{X}_i] \leq D. \quad (41)$$

We will first relax the realism constraint to the weaker constraints below and then show that the solution also satisfies the stronger realism constraint:

$$\mathbb{E}[\tilde{\mathbf{X}}] = 0, \quad \mathbb{E}[\tilde{X}_i^2] = \lambda_i. \quad (42)$$

Consider relaxing the problem even further and jointly optimize over both $\mathbf{X}$ and $\tilde{\mathbf{X}}$ with constraints on the first and second moments of both random variables. The joint maximum entropy distribution then takes the form

$$p(\mathbf{x}, \tilde{\mathbf{x}}) \propto \exp\left(\sum_i \beta_i x_i + \sum_i \gamma_i \tilde{x}_i + \sum_i \mu_i x_i^2 + \sum_i \nu_i \tilde{x}_i^2 + \sum_i \zeta_i x_i \tilde{x}_i\right), \quad (43)$$

that is, it is Gaussian with a precision matrix which has zeros everywhere except the diagonal and the off-diagonals corresponding to the interactions between $X_i$ and $\tilde{X}_i$. In other words, the joint precision matrix $\mathbf{S}$ consists of four blocks where each block is a diagonal matrix. It is not difficult to see by blockwise inversion that then the covariance matrix $\mathbf{C} = \mathbf{S}^{-1}$ must have the same structure. Let $\mathbf{C}_{\mathbf{X}\tilde{\mathbf{X}}}$ be the diagonal matrix corresponding to the covariance between $\mathbf{X}$ and $\tilde{\mathbf{X}}$. We need to maximize

$$h[\mathbf{X}, \tilde{\mathbf{X}}] + \text{const} \propto \ln|\mathbf{C}| \quad (44)$$

$$= \ln|\mathbf{C}_{\mathbf{X}\mathbf{X}}\mathbf{C}_{\tilde{\mathbf{X}}\tilde{\mathbf{X}}} - \mathbf{C}_{\tilde{\mathbf{X}}\mathbf{X}}\mathbf{C}_{\mathbf{X}\tilde{\mathbf{X}}}| \quad (45)$$

$$= \sum_i \ln(\lambda_i^2 - \mathbb{E}[X_i \tilde{X}_i]^2) \quad (46)$$

subject to

$$\sum_i \mathbb{E}[X_i \tilde{X}_i] \geq \sum_i \lambda_i - \frac{D}{2}. \quad (47)$$

Let $c_i = \mathbb{E}[X_i \tilde{X}_i]$ and form the Lagrangian

$$\mathcal{L}(\mathbf{c}, \eta, \mu) = \frac{1}{2}\sum_i \ln(\lambda_i^2 - c_i^2) + \eta\left(\sum_i c_i - \sum_i \lambda_i + \frac{D}{2}\right) + \sum_i \mu_i c_i, \quad (48)$$

where the last term is due to the constraint $c_i \geq 0$. The KKT conditions are

$$\frac{\partial \mathcal{L}}{\partial c_i} = -\frac{c_i}{\lambda_i^2 - c_i^2} + \eta + \mu_i = 0, \quad (49)$$

$$\eta\left(\sum_i c_i - \sum_i \lambda_i + \frac{D}{2}\right) = 0, \qquad\qquad \mu_i c_i = 0, \quad (50)$$

$$\sum_i c_i - \sum_i \lambda_i + \frac{D}{2} \geq 0, \qquad\qquad c_i \geq 0, \quad (51)$$

$$\mu_i \geq 0, \qquad\qquad \eta \geq 0, \quad (52)$$

yielding $c_i = 0$ or

$$c_i = \sqrt{\lambda_i^2 + \frac{1}{4\eta^2}} - \frac{1}{2\eta}. \quad (53)$$

If $c_i = 0$ for some $i$, then $\mu_i = -\eta$ (Eq. 49) and therefore $\mu_i = \eta = 0$ by Eq. 52. But then $c_i = 0$ by Eq. 49 for all $i$. By Eq. 51, we must then have

$$D \geq 2\sum_i \lambda_i. \quad (54)$$

This implies that for sufficiently large distortion, we must have $c_i > 0$ for all $i$. Defining $\theta = (2\eta)^{-1}$ gives that for $\tilde{\mathbf{X}}$ to be optimal, we must have

$$\mathbb{E}[X_i \tilde{X}_i] = c_i = \sqrt{\lambda_i^2 + \theta^2} - \theta \tag{55}$$

for some $\theta$ determined by $D$. Summarizing what we have so far, we have shown that (under relaxed realism constraints) the rate is minimized by a random variable $\tilde{\mathbf{X}}$ jointly Gaussian with $\mathbf{X}$ and a covariance matrix whose entries are zero except those specified by Eqs. 42 and 55. Since the marginal distribution of $\tilde{\mathbf{X}}$ is Gaussian with the desired mean and covariance, it also satisfies the stronger realism constraint $\tilde{\mathbf{X}} \sim \mathbf{X}$.

On the other hand, $\hat{\mathbf{X}} \sim \tilde{\mathbf{X}} \mid \mathbf{X}$. In particular,

$$\mathbb{E}[X_i \hat{X}_i] = \mathbb{E}[X_i Z_i] \tag{56}$$
$$= \mathbb{E}[\alpha_i X_i X_i + \sqrt{1 - \alpha_i^2} \sqrt{\lambda_i} X_i U_i] \tag{57}$$
$$= \alpha_i \mathbb{E}[X_i^2] \tag{58}$$
$$= \alpha_i \lambda_i \tag{59}$$
$$= \sqrt{\lambda_i^2 + \theta^2} - \theta. \tag{60}$$

has the desired property. Thus, $\hat{\mathbf{X}}$ minimizes the rate at any given level of distortion. $\square$

## E    PROOF OF THEOREM 1

**Lemma 2.** *Let $\mathbf{X} : \Omega \to \mathbb{R}^M$ be a random variable with finite differential entropy and let $\mathbf{X}^*$ be a Gaussian random variable with matching first and second-order moments. Let $R(D)$ be the rate-distortion function of $\mathbf{X}$ and $R^*(D)$ be the rate-distortion function of $\mathbf{X}^*$. Then*

$$R^*(D) \le R(D) + D_{\mathrm{KL}}[P_{\mathbf{X}} \parallel P_{\mathbf{X}^*}]. \tag{61}$$

*Proof.* Zamir and Feder (1996) proved the result for $M = 1$. We here extend the proof to $M > 1$. First, observe that

$$D_{\mathrm{KL}}[P_{\mathbf{X}} \parallel P_{\mathbf{X}^*}] = \mathbb{E}[-\log p_{\mathbf{X}^*}(\mathbf{X})] - h[\mathbf{X}] \tag{62}$$
$$= \mathbb{E}[-\log p_{\mathbf{X}^*}(\mathbf{X}^*)] - h[\mathbf{X}] \tag{63}$$
$$= h[\mathbf{X}^*] - h[\mathbf{X}] \tag{64}$$

since $\log p_{\mathbf{X}^*}$ is a quadratic form and $\mathbf{X}$ and $\mathbf{X}^*$ have matching moments. By the Shannon lower bound (Wu, 2016),

$$R(D) \ge h[\mathbf{X}] - \frac{M}{2} \log\left(2\pi e \frac{D}{M}\right). \tag{65}$$

Let $\lambda_1, \dots, \lambda_M$ be the eigenvalues of the covariance of $\mathbf{X}$. Then by Eqs. 64 and 65 we have

$$R(D) \ge h[\mathbf{X}^*] - D[P_{\mathbf{X}}, P_{\mathbf{X}^*}] - \frac{M}{2} \log\left(2\pi e \frac{D}{M}\right) \tag{66}$$
$$= \sum_i \frac{1}{2} \log\left(\frac{\lambda_i}{D/M}\right) - D[P_{\mathbf{X}}, P_{\mathbf{X}^*}] \tag{67}$$
$$\ge \sum_i \frac{1}{2} \log\left(\frac{\lambda_i}{D_i}\right) - D[P_{\mathbf{X}}, P_{\mathbf{X}^*}] \tag{68}$$
$$= R^*(D) - D[P_{\mathbf{X}}, P_{\mathbf{X}^*}]. \tag{69}$$

where $D_i = \min(\theta, \lambda_i)$ and $\theta$ is such that $D = \sum_i D_i$. The inequality follows from the optimality of the water-filling solution given by the $D_i$ (Shannon, 1949; Cover and Thomas, 2006). Bringing the KL divergence to the other side of the equation gives the desired result. $\square$

**Theorem 1.** *Let $\mathbf{X} : \Omega \to \mathbb{R}^M$ be a random variable with finite differential entropy, zero mean and covariance* $\mathrm{diag}(\lambda_1, \ldots, \lambda_M)$. *Let $\mathbf{U} \sim \mathcal{N}(0, \mathbf{I})$ and define*

$$Z_i = \sqrt{1 - \gamma_i^2} X_i + \gamma_i \sqrt{\lambda_i} U_i, \qquad\qquad \hat{\mathbf{X}} \sim P(\mathbf{X} \mid \mathbf{Z}). \tag{70}$$

*where $\gamma_i^2 = \min(1, \theta/\lambda_i)$ for some $\theta$. Further, let $\mathbf{X}^*$ be a Gaussian random variable with the same first and second-order moments as $\mathbf{X}$ and let $\mathbf{Z}^*$ be defined analogously to $\mathbf{Z}$ but in terms of $\mathbf{X}^*$. Then if $R$ is the rate-distortion function of $\mathbf{X}$ and $R^*$ is the rate-distortion function of $\mathbf{X}^*$,*

$$I[\mathbf{X}, \mathbf{Z}] \le R^*(D/2) - D_{\mathrm{KL}}[P_{\mathbf{Z}} \parallel P_{\mathbf{Z}^*}] \tag{71}$$
$$\le R(D/2) + D_{\mathrm{KL}}[P_{\mathbf{X}} \parallel P_{\mathbf{X}^*}] - D_{\mathrm{KL}}[P_{\mathbf{Z}} \parallel P_{\mathbf{Z}^*}] \tag{72}$$

*where $D = \mathbb{E}[\|\mathbf{X} - \hat{\mathbf{X}}\|^2]$.*

*Proof.* We have

$$D_i = \mathbb{E}[(X_i - \hat{X}_i)^2] \tag{73}$$
$$= \mathbb{E}[(X_i - \mathbb{E}[X_i \mid \mathbf{Z}] + \mathbb{E}[X_i \mid \mathbf{Z}] - \hat{X}_i)^2] \tag{74}$$
$$= \mathbb{E}[(X_i - \mathbb{E}[X_i \mid \mathbf{Z}])^2 + (\mathbb{E}[X_i \mid \mathbf{Z}] - \hat{X}_i)^2 - 2(X_i - \mathbb{E}[X_i \mid \mathbf{Z}])(\mathbb{E}[X_i \mid \mathbf{Z}] - \hat{X}_i)] \tag{75}$$
$$= \mathbb{E}[(X_i - \mathbb{E}[X_i \mid \mathbf{Z}])^2] + \mathbb{E}[(\mathbb{E}[\hat{X}_i \mid \mathbf{Z}] - \hat{X}_i)^2] \tag{76}$$
$$\quad - 2\mathbb{E}_{\mathbf{Z}}[\mathbb{E}_{X_i}[\overline{X_i - \mathbb{E}[X_i \mid \mathbf{Z}] \mid \mathbf{Z}}] \mathbb{E}_{\hat{X}_i}[\mathbb{E}[X_i \mid \mathbf{Z}] - \hat{X}_i \mid \mathbf{Z}]] \tag{77}$$
$$= 2\mathbb{E}[(X_i - \mathbb{E}[X_i \mid \mathbf{Z}])^2] \tag{78}$$
$$\le 2\mathbb{E}[(X_i - \sqrt{1 - \gamma_i^2} Z_i)^2] \tag{79}$$
$$= 2\mathbb{E}[(1 - (1 - \gamma_i^2))X_i - \sqrt{1 - \gamma_i^2}\gamma_i\sqrt{\lambda_i}U_i)^2] \tag{80}$$
$$= 2(\mathrm{Var}[\gamma_i^2 X_i] + \mathrm{Var}[\sqrt{1 - \gamma_i^2}\gamma_i\sqrt{\lambda_i}U_i]) \tag{81}$$
$$= 2(\gamma_i^4 \lambda_i + (1 - \gamma_i^2)\gamma_i^2 \lambda_i) \tag{82}$$
$$= 2\gamma_i^2 \lambda_i, \tag{83}$$

where in Eq. 77 we used that $\mathbf{X} \perp\!\!\!\perp \hat{\mathbf{X}} \mid \mathbf{Z}$ and in Eq. 78 we used that $(\mathbf{X}, \mathbf{Z}) \sim (\hat{\mathbf{X}}, \mathbf{Z})$. Eq. 79 follows because the conditional expectation minimizes the squared error among all estimators of $X_i$. For the overall distortion, we therefore have

$$D = \mathbb{E}[\|\mathbf{X} - \hat{\mathbf{X}}\|^2] = \sum_i D_i \le 2 \sum_i \gamma_i^2 \lambda_i = D^*. \tag{84}$$

Note that $D^*$ is the distortion we would have gotten if $\mathbf{X}$ were Gaussian (Eq. 14). Define

$$V_i = (1 - \gamma_i^2)^{-\frac{1}{2}} \gamma_i \sqrt{\lambda_i} U_i \quad \text{and} \quad Y_i = (1 - \gamma_i^2)^{-\frac{1}{2}} Z_i = X_i + V_i \tag{85}$$

and let $\mathbf{Y}^*$ be the Gaussian random variable defined analogously to $\mathbf{Y}$ except in terms of $\mathbf{Z}^*$ instead of $\mathbf{Z}$. To obtain the rate, first observe that

$$I[\mathbf{X}^*, \mathbf{Z}^*] = I[\mathbf{X}^*, \mathbf{Y}^*] \tag{86}$$
$$= I[\mathbf{X}^*, \mathbf{X}^* + \mathbf{V}] \tag{87}$$
$$= h[\mathbf{X}^* + \mathbf{V}] - h[\mathbf{X}^* + \mathbf{V} \mid \mathbf{X}^*] \tag{88}$$
$$= h[\mathbf{X}^* + \mathbf{V}] - h[\mathbf{V}] \tag{89}$$
$$= h[\mathbf{X} + \mathbf{V}] - h[\mathbf{V}] + h[\mathbf{X}^* + \mathbf{V}] - h[\mathbf{X} + \mathbf{V}] \tag{90}$$
$$= h[\mathbf{X} + \mathbf{V}] - h[\mathbf{X} + \mathbf{V} \mid \mathbf{X}] + h[\mathbf{X}^* + \mathbf{V}] - h[\mathbf{X} + \mathbf{V}] \tag{91}$$
$$= I[\mathbf{X}, \mathbf{X} + \mathbf{V}] - \mathbb{E}[\log p_{\mathbf{Y}^*}(\mathbf{Y}^*)] + \mathbb{E}[\log p_{\mathbf{Y}}(\mathbf{Y})] \tag{92}$$
$$= I[\mathbf{X}, \mathbf{Y}] - \mathbb{E}[\log p_{\mathbf{Y}^*}(\mathbf{Y}^*)] + \mathbb{E}[\log p_{\mathbf{Y}}(\mathbf{Y})] \tag{93}$$
$$= I[\mathbf{X}, \mathbf{Y}] - \mathbb{E}[\log p_{\mathbf{Y}^*}(\mathbf{Y})] + \mathbb{E}[\log p_{\mathbf{Y}}(\mathbf{Y})] \tag{94}$$
$$= I[\mathbf{X}, \mathbf{Y}] + D_{\mathrm{KL}}[P_{\mathbf{Y}} \parallel P_{\mathbf{Y}^*}] \tag{95}$$
$$= I[\mathbf{X}, \mathbf{Z}] + D_{\mathrm{KL}}[P_{\mathbf{Z}} \parallel P_{\mathbf{Z}^*}]. \tag{96}$$

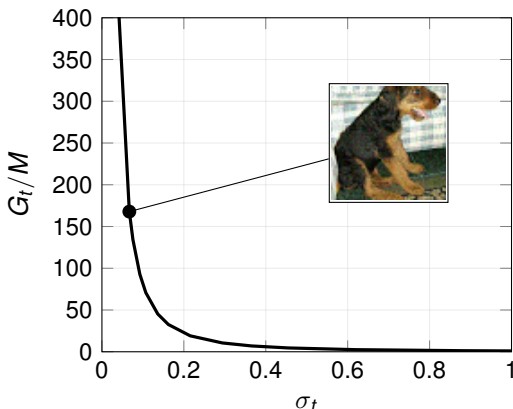

Figure 5: While $G_0$ is undefined for images with discretized pixels, we may instead consider the distribution of pixels with imperceptible Gaussian noise added to it. We can estimate the corresponding $G_t$ using the diffusion model, the results of which are shown in this plot. $G_t$ converges to $M$ as $\sigma_t$ approaches 1.

where the first step and last step follow from the invariance of mutual information and KL divergence under invertible transformations. Eq. 94 follows because $\log p_{\mathbf{Y}^*}$ is a quadratic form and $\mathbf{Y}$ and $\mathbf{Y}^*$ have matching moments. We therefore have

$$I[\mathbf{X}, \mathbf{Z}] = I[\mathbf{X}^*, \mathbf{Z}^*] - D_{\mathrm{KL}}[P_{\mathbf{Z}} \,||\, P_{\mathbf{Z}^*}] \tag{97}$$

$$= R^*(D^*/2) - D_{\mathrm{KL}}[P_{\mathbf{Z}} \,||\, P_{\mathbf{Z}^*}] \tag{98}$$

$$\leq R^*(D/2) - D_{\mathrm{KL}}[P_{\mathbf{Z}} \,||\, P_{\mathbf{Z}^*}] \tag{99}$$

$$\leq R(D/2) + D_{\mathrm{KL}}[P_{\mathbf{X}} \,||\, P_{\mathbf{X}^*}] - D_{\mathrm{KL}}[P_{\mathbf{Z}} \,||\, P_{\mathbf{Z}^*}], \tag{100}$$

where the second equality is due to Eq. 14, the first inequality is due to $D \leq D^*$, and the second inequality follows from the Shannon lower bound and Lemma 2. $\qquad\square$

## F  PROOF OF THEOREM 2

Theorem 2 compares the reconstruction error of $\hat{\mathbf{X}}_A \sim P(\mathbf{X} \,|\, \mathbf{Z}_t)$ with $\hat{\mathbf{X}}_F = \hat{\mathbf{Z}}_0$ where $\hat{\mathbf{Z}}_0$ is the solution of

$$d\mathbf{z}_t = \left( -\frac{1}{2}\beta_t \mathbf{z}_t - \frac{1}{2}\beta_t \nabla \ln p_t(\mathbf{z}_t) \right) dt. \tag{101}$$

given $\mathbf{Z}_t = \sqrt{1 + \sigma_t^2}\mathbf{X} + \sigma_t \mathbf{U}$ as initial condition. To derive the following results it is often more convenient to work with the "variance exploding" diffusion process

$$\mathbf{Y}_t = (1 - \sigma_t^2)^{-\frac{1}{2}}\mathbf{Z}_t \sim \mathbf{X} + (1 - \sigma_t^2)^{-\frac{1}{2}}\sigma_t \mathbf{U} = \mathbf{X} + \eta_t \mathbf{U} \tag{102}$$

instead of the "variance preserving" process $\mathbf{Z}_t$ (Song et al., 2021). We use $\tilde{p}_t$ for the marginal density of $\mathbf{Y}_t$ and reserve $p_t$ for the density of $\mathbf{Z}_t$.

We further define the following quantity for continuously differentiable densities, which can be viewed as a measure of smoothness of a density,

$$G = \mathbb{E}[\|\nabla \ln p(\mathbf{X})\|^2]. \tag{103}$$

It can be shown that when $\mathbb{E}[\|\mathbf{X}\|^2] = 1$, we have $G \geq M$ with equality when $\mathbf{X}$ is isotropic Gaussian. That is, the isotropic Gaussian is the smoothest distribution (with a continuously differentiable density) as measured by $G$. We further define

$$G_t = \mathbb{E}[\|\nabla_{\mathbf{z}} \log p_t(\mathbf{Z}_t)\|^2] \quad \text{and} \quad \tilde{G}_t = \mathbb{E}[\|\nabla_{\mathbf{y}} \log \tilde{p}_t(\mathbf{Y}_t)\|^2] \tag{104}$$

which are linked by the chain rule, $\tilde{G}_t = (1 - \sigma_t^2)G_t$. Using a trained diffusion model, we can obtain estimates of $G_t$ for ImageNet 64x64, which are shown in Fig. 5.

**Lemma 3.** *Diffusion increases the smoothness of a distribution, $\tilde{G}_t \leq G_0$.*

*Proof.* We have

$$\nabla_{\mathbf{y}} \ln \tilde{p}_t(\mathbf{y}_t) = \int p(\mathbf{u} \mid \mathbf{y}_t) \nabla_{\mathbf{y}} \ln \tilde{p}_t(\mathbf{y}) \, d\mathbf{u} \tag{105}$$

$$= \int p(\mathbf{u} \mid \mathbf{y}_t) \nabla_{\mathbf{y}} \ln \frac{\tilde{p}_t(\mathbf{y}) p(\mathbf{u} \mid \mathbf{y}_t)}{p(\mathbf{u} \mid \mathbf{y}_t)} \, d\mathbf{u} \tag{106}$$

$$= \int p(\mathbf{u} \mid \mathbf{y}_t) \nabla_{\mathbf{y}} \ln p(\mathbf{u}) p(\mathbf{y}_t \mid \mathbf{u}) \, d\mathbf{u} - \int p(\mathbf{u} \mid \mathbf{y}_t) \nabla_{\mathbf{y}} \ln p(\mathbf{u} \mid \mathbf{y}_t) \, d\mathbf{u} \tag{107}$$

$$= \int p(\mathbf{u} \mid \mathbf{y}_t) \nabla_{\mathbf{y}} \ln p(\mathbf{y}_t \mid \mathbf{u}) \, d\mathbf{x} - 0 \tag{108}$$

$$= \int p(\mathbf{u} \mid \mathbf{y}_t) \nabla_{\mathbf{y}} \ln p_0(\mathbf{y}_t - \eta_t \mathbf{u}) \, d\mathbf{x} \tag{109}$$

and therefore

$$\tilde{G}_t = \mathbb{E}[\|\nabla_{\mathbf{y}} \ln \tilde{p}_t(\mathbf{Y}_t)\|^2] \tag{110}$$

$$= \mathbb{E}[\|\mathbb{E}[\nabla_{\mathbf{y}} \ln p_0(\mathbf{Y}_t - \eta_t \mathbf{U}) \mid \mathbf{Y}_t]\|^2] \tag{111}$$

$$\leq \mathbb{E}[\mathbb{E}[\|\nabla_{\mathbf{y}} \ln p_0(\mathbf{Y}_t - \eta_t \mathbf{U})\|^2 \mid \mathbf{Y}_t]] \tag{112}$$

$$= \mathbb{E}[\|\nabla_{\mathbf{x}} \ln p_0(\mathbf{X})\|^2] \tag{113}$$

$$= G_0 \tag{114}$$

due to Jensen's inequality. $\square$

We will also need the following known useful identity which is a special case of Tweedie's formula (Robbins, 1956). We include a derivation for completeness.

**Lemma 4.** *Let $\mathbf{X}$ have a density and let $\mathbf{Y}_t$, $\tilde{p}_t$, and $\eta_t$ be defined as above. Then*

$$\mathbb{E}[\mathbf{X} \mid \mathbf{y}_t] = \mathbf{y}_t + \eta_t^2 \nabla \ln \tilde{p}_t(\mathbf{y}_t). \tag{115}$$

*Proof.*

$$\nabla_{\mathbf{y}} \log \tilde{p}_t(\mathbf{y}_t) = \int p(\mathbf{x} \mid \mathbf{y}_t) \nabla_{\mathbf{y}} \log \tilde{p}_t(\mathbf{y}_t) \, d\mathbf{x} \tag{116}$$

$$= \int p(\mathbf{x} \mid \mathbf{y}_t) \nabla_{\mathbf{y}} \log \frac{\tilde{p}_t(\mathbf{y}_t) p(\mathbf{x} \mid \mathbf{y}_t)}{p(\mathbf{x} \mid \mathbf{y}_t)} \, d\mathbf{x} \tag{117}$$

$$= \int p(\mathbf{x} \mid \mathbf{y}_t) \nabla_{\mathbf{y}} \log \left( p(\mathbf{y}_t \mid \mathbf{x}) p(\mathbf{x}) \right) \, d\mathbf{x} \tag{118}$$

$$\qquad - \int p(\mathbf{x} \mid \mathbf{y}_t) \nabla_{\mathbf{y}} \log p(\mathbf{x} \mid \mathbf{y}_t) \, d\mathbf{x} \tag{119}$$

$$= \int p(\mathbf{x} \mid \mathbf{y}_t) \nabla_{\mathbf{y}} \log p(\mathbf{y}_t \mid \mathbf{x}) \, d\mathbf{x} \tag{120}$$

$$= \int p(\mathbf{x} \mid \mathbf{y}_t) \nabla_{\mathbf{y}} \log \mathcal{N} \left( \mathbf{y}_t; \mathbf{x}, \eta_t^2 \mathbf{I} \right) \, d\mathbf{x} \tag{121}$$

$$= \int p(\mathbf{x} \mid \mathbf{y}_t) \frac{1}{\eta_t^2} (\mathbf{x} - \mathbf{y}_t) \, d\mathbf{x} \tag{122}$$

$$= \frac{1}{\eta_t^2} \left( \mathbb{E}[\mathbf{X} \mid \mathbf{y}_t] - \mathbf{y}_t \right) \tag{123}$$

$\square$

The following three lemmas relate the reconstruction errors of $\hat{\mathbf{X}}_A$ and $\hat{\mathbf{X}}_F$ to the smoothness of the source distribution as measured by $G_0$ and $G_t$.

**Lemma 5.** *Let* $\mathbf{X}$ *have a density and let* $\hat{\mathbf{X}}_A$, $\eta_t$, *and* $G_t$ *be defined as above. Then*

$$\mathbb{E}[\|\hat{\mathbf{X}}_A - \mathbf{X}\|^2] = 2\eta_t^2 M - 2\eta_t^4 \tilde{G}_t \tag{124}$$
$$= 2\eta_t^2 M - 2\eta_t^4 (1 - \sigma_t^2) G_t \tag{125}$$

*Proof.*

$$\mathbb{E}[\|\hat{\mathbf{X}}_A - \mathbf{X}\|^2] = \mathbb{E}[\|\hat{\mathbf{X}}_A - \mathbf{Y}_t + \mathbf{Y}_t - \mathbf{X}\|^2] \tag{126}$$
$$= \mathbb{E}[\|\hat{\mathbf{X}}_A - \mathbf{Y}_t\|^2 + \|\mathbf{Y}_t - \mathbf{X}\|^2 + 2(\hat{\mathbf{X}}_A - \mathbf{Y}_t)^\top(\mathbf{Y}_t - \mathbf{X})] \tag{127}$$
$$= \mathbb{E}[\|\hat{\mathbf{X}}_A - \mathbf{Y}_t\|^2] + \mathbb{E}[\|\mathbf{Y}_t - \mathbf{X}\|^2] \tag{128}$$
$$\quad + 2\mathbb{E}[\mathbb{E}[\hat{\mathbf{X}}_A - \mathbf{Y}_t \mid \mathbf{Y}_t]^\top \mathbb{E}[\mathbf{Y}_t - \mathbf{X} \mid \mathbf{Y}_t]] \tag{129}$$
$$= \mathbb{E}[\|\mathbf{X} - \mathbf{Y}_t\|^2] + \mathbb{E}[\|\mathbf{Y}_t - \mathbf{X}\|^2] \tag{130}$$
$$\quad + 2\mathbb{E}[\mathbb{E}[\mathbf{X} - \mathbf{Y}_t \mid \mathbf{Y}_t]^\top \mathbb{E}[\mathbf{Y}_t - \mathbf{X} \mid \mathbf{Y}_t]] \tag{131}$$
$$= 2\mathbb{E}[\|\mathbf{Y}_t - \mathbf{X}\|^2] - 2\mathbb{E}[\|\mathbb{E}[\mathbf{Y}_t - \mathbf{X} \mid \mathbf{Y}_t]\|^2] \tag{132}$$
$$= 2\mathbb{E}[\|\eta_t \mathbf{U}_t\|^2] - 2\mathbb{E}[\|\mathbf{Y}_t - \mathbb{E}[\mathbf{X} \mid \mathbf{Y}_t]\|^2] \tag{133}$$
$$= 2\eta_t^2 M - 2\mathbb{E}[\|\eta_t^2 \nabla_{\mathbf{y}} \ln \tilde{p}_t(\mathbf{Y}_t)\|^2] \tag{134}$$
$$= 2\eta_t^2 M - 2\mathbb{E}[\|\eta_t^2 (1 - \sigma_t^2)^{\frac{1}{2}} \nabla_{\mathbf{z}} \ln p_t(\mathbf{Z}_t)\|^2] \tag{135}$$
$$= 2\eta_t^2 M - 2\eta_t^4 (1 - \sigma_t^2) G_t \tag{136}$$

$\square$

**Lemma 6.** *Let* $\mathbf{X}$ *have a density and let* $\hat{\mathbf{X}}_F$, $\eta_t$, *and* $G_0$ *be defined as above. Then*

$$\mathbb{E}[\|\hat{\mathbf{X}}_F - \mathbf{Y}_t\|^2] \leq \frac{1}{4}\eta_t^4 G_0. \tag{137}$$

*Proof.* We have

$$\mathbb{E}[\|\hat{\mathbf{X}}_F - \mathbf{Y}_t\|^2] = \mathbb{E}[\|F_t^{-1}(\mathbf{Y}_t) - \mathbf{Y}_t\|^2] = \mathbb{E}[\|\mathbf{X} - F_t(\mathbf{X})\|^2] \tag{138}$$

where $F_t$ is the invertible function which maps $\mathbf{x}$ to $\mathbf{y}_t$ according to the ODE

$$d\mathbf{y}_t = -\alpha_t \nabla \ln p_t(\mathbf{y}_t)\,dt, \quad \mathbf{y}_0 = \mathbf{x}, \tag{139}$$

where $\alpha_t$ relates to $\eta_t$ as follows,

$$\eta_t = \sqrt{2\int_0^t \alpha_\tau\,d\tau}. \tag{140}$$

Different schedules are equivalent up to reparametrization of the time parameter (Kingma et al., 2021). For now, assume the parametrization $\alpha_t = 1$ (or $\eta_t = \sqrt{2t}$). Integrating the above ODE then yields

$$\mathbf{y}_t = F_t(\mathbf{x}) = \mathbf{x} - \int_0^t \nabla \ln p_\tau(\mathbf{y}_\tau)\,d\tau. \tag{141}$$

Consider the following Riemann sum approximation of $F_t$,

$$F_{t,N}(\mathbf{x}) = \mathbf{x} - \sum_{n=0}^{N-1} \frac{t}{N}\nabla \ln p_{t_n}(\mathbf{y}_{t_n}) \tag{142}$$

where $t_n = nt/N$ and $\mathbf{y}_{t_n} = F_{t_n}(\mathbf{x})$. Since the gradient of the log-density is continuous and the integral is over a compact interval, the partial derivatives are bounded inside the interval and the Riemann sum converges to

$$F_t(\mathbf{x}) = \lim_{N\to\infty} F_{t,N}(\mathbf{x}). \tag{143}$$

Thus,

$$\mathbb{E}[\|\mathbf{X} - F_t(\mathbf{X})\|^2] = \mathbb{E}[\|\mathbf{X} - \lim_{N \to \infty} F_{t,N}(\mathbf{X})\|^2] \tag{144}$$

$$= \mathbb{E}\left[\left\|\lim_{N \to \infty} \sum_{n=0}^{N-1} \frac{t}{N} \nabla \ln \tilde{p}_{t_n}(\mathbf{Y}_{t_n})\right\|^2\right] \tag{145}$$

$$= \mathbb{E}\left[\lim_{N \to \infty} \left\|\sum_{n=0}^{N-1} \frac{t}{N} \nabla \ln \tilde{p}_{t_n}(\mathbf{Y}_{t_n})\right\|^2\right] \tag{146}$$

$$\leq \mathbb{E}\left[\lim_{N \to \infty} \sum_{n=0}^{N-1} \frac{1}{N} \|t \nabla \ln \tilde{p}_{t_n}(\mathbf{Y}_{t_n})\|^2\right] \tag{147}$$

$$= \lim_{N \to \infty} \frac{t^2}{N} \sum_{n=0}^{N-1} \mathbb{E}\left[\|\nabla \ln \tilde{p}_{t_n}(\mathbf{Y}_{t_n})\|^2\right] \tag{148}$$

$$= \lim_{N \to \infty} \frac{t^2}{N} \sum_{n=0}^{N-1} \tilde{G}_{t_n} \tag{149}$$

$$\leq \lim_{N \to \infty} \frac{t^2}{N} \sum_{n=0}^{N-1} G_0 \tag{150}$$

$$= t^2 G_0 \tag{151}$$

$$= \frac{1}{4} \eta_t^4 G_0, \tag{152}$$

Eq. 147 again uses Jensen's inequality. Eq. 148 (swapping limit and expectation) follows from the dominated convergence theorem since each element of the sequence is bounded by $t^2 G_0$ (Lemma 3). $\square$

**Lemma 7.** *Let* $\mathbf{X}$ *have a smooth density and let* $\hat{\mathbf{X}}_F$, $\eta_t$, *and* $G_0$ *be defined as above. Then*

$$\mathbb{E}[\|\hat{\mathbf{X}}_F - \mathbf{X}\|^2] \leq \eta_t^2 M + \frac{1}{2} \eta_t^4 G_0 + 2\eta_t^4 (1 - \sigma_t^2) G_t. \tag{153}$$

*Proof.*

$$\mathbb{E}[\|\hat{\mathbf{X}}_F - \mathbf{X}\|^2] = \mathbb{E}[\|\hat{\mathbf{X}}_F - \mathbb{E}[\mathbf{X} \mid \mathbf{Y}_t] + \mathbb{E}[\mathbf{X} \mid \mathbf{Y}_t] - \mathbf{X}\|^2] \tag{154}$$

$$= \mathbb{E}[\|\hat{\mathbf{X}}_F - \mathbb{E}[\mathbf{X} \mid \mathbf{Y}_t]\|^2] + \mathbb{E}[\|\mathbb{E}[\mathbf{X} \mid \mathbf{Y}_t] - \mathbf{X}\|^2] + 0 \tag{155}$$

$$\leq \mathbb{E}[\|\hat{\mathbf{X}}_F - \mathbb{E}[\mathbf{X} \mid \mathbf{Y}_t]\|^2] + \mathbb{E}[\|\mathbf{Y}_t - \mathbf{X}\|^2] \tag{156}$$

$$= \mathbb{E}\left[\|\hat{\mathbf{X}}_F - \mathbf{Y}_t + \eta_t^2 \nabla \ln \tilde{p}_t(\mathbf{Y}_t)\|^2\right] + \mathbb{E}[\|\eta_t \mathbf{U}\|^2] \tag{157}$$

$$= \mathbb{E}\left[4 \left\|\frac{1}{2}(\hat{\mathbf{X}}_F - \mathbf{Y}_t) + \frac{1}{2}\left(\eta_t^2 \nabla \ln \tilde{p}_t(\mathbf{Y}_t)\right)\right\|^2\right] + \eta_t^2 M \tag{158}$$

$$\leq 2\mathbb{E}[\|\hat{\mathbf{X}}_F - \mathbf{Y}_t\|^2] + 2\mathbb{E}[\|\eta_t^2 \nabla \ln \tilde{p}_t(\mathbf{Y}_t)\|^2] + \eta_t^2 M \tag{159}$$

$$= 2\mathbb{E}[\|\hat{\mathbf{X}}_F - \mathbf{Y}_t\|^2] + 2\eta_t^4 (1 - \sigma_t^2) G_t + \eta_t^2 M \tag{160}$$

$$\leq \frac{1}{2} \eta_t^4 G_0 + 2\eta_t^4 (1 - \sigma_t^2) G_t + \eta_t^2 M \tag{161}$$

where the first inequality is due to the conditional expectation minimizing squared error, the second inequality is due to Jensen's inequality and the last inequality is due to Lemma 6. $\square$

We are finally in a position to prove Theorem 2.

**Theorem 2.** *Let* $\mathbf{X} : \Omega \to \mathbb{R}^M$ *have a smooth density* $p$ *with finite*

$$G = \mathbb{E}[\|\nabla \ln p(\mathbf{X})\|^2]. \tag{162}$$

*Let $\mathbf{Z}_t = \sqrt{1 - \sigma_t^2}\mathbf{X} + \sigma_t\mathbf{U}$ with $\mathbf{U} \sim \mathcal{N}(0, \mathbf{I})$. Let $\hat{\mathbf{X}}_A \sim P(\mathbf{X} \mid \mathbf{Z}_t)$ and let $\hat{\mathbf{X}}_F = \hat{\mathbf{Z}}_0$ be the solution to Eq. 6 with $\mathbf{Z}_t$ as initial condition. Then*

$$\lim_{\sigma_t \to 0} \frac{\mathbb{E}[\|\hat{\mathbf{X}}_F - \mathbf{X}\|^2]}{\mathbb{E}[\|\hat{\mathbf{X}}_A - \mathbf{X}\|^2]} = \frac{1}{2} \tag{163}$$

*Proof.* The limit is to be understood as the one-sided limit from above. We have

$$\lim_{\sigma_t \to 0} \frac{\mathbb{E}[\|\hat{\mathbf{X}}_F - \mathbf{X}\|^2]}{\mathbb{E}[\|\hat{\mathbf{X}}_A - \mathbf{X}\|^2]} \leq \lim_{\sigma_t \to 0} \frac{\eta_t^2 M + \frac{1}{2}\eta_t^4 G_0 + 2\eta_t^4(1 - \sigma_t^2)G_t}{2\eta_t^2 M - 2\eta_t^4(1 - \sigma_t^2)G_t} \tag{164}$$

$$\leq \lim_{\sigma_t \to 0} \frac{\eta_t^2 M + \frac{1}{2}\eta_t^4 G_0 + 2\eta_t^4 G_0}{2\eta_t^2 M - 2\eta_t^4 G_0} \tag{165}$$

$$= \lim_{\eta_t \to 0} \frac{2\eta_t M + 2\eta_t^3 G_0 + 8\eta_t^3 G_0}{4\eta_t M - 8\eta_t^3 G_0} \tag{166}$$

$$= \lim_{\eta_t \to 0} \frac{2M + 6\eta_t^2 G_0 + 24\eta_t^2 G_0}{4M - 24\eta_t^2 G_0} \tag{167}$$

$$= \frac{2M}{4M} \tag{168}$$

$$= \frac{1}{2} \tag{169}$$

where the first inequality follows from Lemmas 5 and 7, the second inequality is due to Lemma 3, and we applied L'Hôpital's rule twice. $\qquad\square$

## G  PROOF OF THEOREM 3

**Theorem 3.** *Let $\mathbf{X} = \mathbf{QS}$ where $\mathbf{Q}$ is an orthogonal matrix and $\mathbf{S} : \Omega \to \mathbb{R}^M$ is a random vector with smooth density and $S_i \perp\!\!\!\perp S_j$ for all $i \neq j$. Define*

$$\mathbf{Z}_t = \sqrt{1 - \sigma_t^2}\mathbf{X} + \sigma_t\mathbf{U} \quad \text{where} \quad \mathbf{U} \sim \mathcal{N}(0, \mathbf{I}). \tag{170}$$

*If $\hat{\mathbf{X}}_F = \hat{\mathbf{Z}}_0$ is the solution to the ODE in Eq. 6 given $\mathbf{Z}_t$ as initial condition, then*

$$\mathbb{E}[\|\hat{\mathbf{X}}_F - \mathbf{X}\|^2] \leq \mathbb{E}[\|\hat{\mathbf{X}}' - \mathbf{X}\|^2] \tag{171}$$

*for any $\hat{\mathbf{X}}'$ with $\hat{\mathbf{X}}' \perp\!\!\!\perp \mathbf{X} \mid \mathbf{Z}_t$ which achieves perfect realism, $\hat{\mathbf{X}}' \sim \mathbf{X}$.*

*Proof.* Define the variance exploding diffusion process as

$$d\mathbf{Y}_t = \sqrt{\zeta_t}d\mathbf{W}_t \quad \text{with} \quad (1 - \sigma_t^2)^{-\frac{1}{2}}\sigma_t^2 = \int_0^t \zeta_\tau \, d\tau \tag{172}$$

so that

$$\mathbf{Y}_t = (1 - \sigma_t^2)^{-\frac{1}{2}}\mathbf{Z}_t \sim \mathbf{X} + (1 - \sigma_t^2)^{-\frac{1}{2}}\sigma_t^2\mathbf{U} = \mathbf{X} + \eta_t\mathbf{U}. \tag{173}$$

Further define $F_t$ as the function which maps $\mathbf{x}$ to the solution of the ODE in Eq. 6 with starting condition $\mathbf{z}_0 = \mathbf{x}$. Then $F_t$ is invertible (Song et al., 2021) and we can write $\hat{\mathbf{X}}_F = F_t^{-1}(\mathbf{Z}_t)$. Further, let $\tilde{F}_t$ be the corresponding function for the variance exploding process such that

$$\tilde{F}_t^{-1}(\mathbf{y}) = F_t^{-1}\left(\sqrt{1 - \sigma_t^2}\mathbf{y}\right), \quad \hat{\mathbf{X}}_F = \tilde{F}_t^{-1}(\mathbf{Y}_t). \tag{174}$$

For arbitrary $\hat{\mathbf{X}}$ with $\hat{\mathbf{X}} \perp\!\!\!\perp \mathbf{X} \mid \mathbf{Y}_t$, we have

$$\mathbb{E}[\|\hat{\mathbf{X}} - \mathbf{X}\|^2] = \mathbb{E}[\|\hat{\mathbf{X}} - \mathbb{E}[\mathbf{X} \mid \mathbf{Y}_t] + \mathbb{E}[\mathbf{X} \mid \mathbf{Y}_t] - \mathbf{X}\|^2] \tag{175}$$

$$= \mathbb{E}[\|\hat{\mathbf{X}} - \mathbb{E}[\mathbf{X} \mid \mathbf{Y}_t]\|^2] + \mathbb{E}[\|\mathbb{E}[\mathbf{X} \mid \mathbf{Y}_t] - \mathbf{X}\|^2] \tag{176}$$

$$+ \mathbb{E}[(\hat{\mathbf{X}} - \mathbb{E}[\mathbf{X} \mid \mathbf{Y}_t])^\top(\mathbb{E}[\mathbf{X} \mid \mathbf{Y}_t] - \mathbf{X})] \tag{177}$$

$$= \mathbb{E}[\|\hat{\mathbf{X}} - \mathbb{E}[\mathbf{X} \mid \mathbf{Y}_t]\|^2] + \mathbb{E}[\|\mathbb{E}[\mathbf{X} \mid \mathbf{Y}_t] - \mathbf{X}\|^2] \tag{178}$$

$$+ \mathbb{E}_{\mathbf{Y}_t}[\mathbb{E}_{\hat{\mathbf{X}}}[\hat{\mathbf{X}} - \mathbb{E}[\mathbf{X} \mid \mathbf{Y}_t] \mid \mathbf{Y}_t]^\top \mathbb{E}_{\mathbf{X}}[\mathbb{E}[\mathbf{X} \mid \mathbf{Y}_t] - \mathbf{X} \mid \mathbf{Y}_t]] \tag{179}$$

$$= \mathbb{E}[\|\hat{\mathbf{X}} - \mathbb{E}[\mathbf{X} \mid \mathbf{Y}_t]\|^2] + \mathbb{E}[\|\mathbb{E}[\mathbf{X} \mid \mathbf{Y}_t] - \mathbf{X}\|^2] \tag{180}$$

Define $\hat{\mathbf{X}}_{\text{MSE}} = \psi_t(\mathbf{Y}_t) = \mathbb{E}[\mathbf{X} \mid \mathbf{Y}_t]$.

Assume $M = 1$ so that $X = S$. We first show that then $\psi_t$ is a monotone function of $y_t$:

$$\psi'(y_t) = \frac{\partial}{\partial y} \mathbb{E}[X \mid y_t] \tag{181}$$

$$= \frac{\partial}{\partial y} \left( y_t + \eta_t^2 \frac{\partial}{\partial y} \ln \tilde{p}_t(y_t) \right) \tag{182}$$

$$= 1 + \eta_t^2 \frac{\partial^2}{\partial y^2} \ln \tilde{p}_t(y_t) \tag{183}$$

$$= 1 + \eta_t^2 \int p(x \mid y_t) \frac{\partial^2}{\partial y^2} \ln \frac{\tilde{p}_t(y_t) p(x \mid y_t)}{p(x \mid y_t)} \, dx \tag{184}$$

$$= 1 + \eta_t^2 \int p(x \mid y_t) \frac{\partial^2}{\partial y^2} \ln \tilde{p}_t(y_t \mid x) \, dx - \eta_t^2 \int p(x \mid y_t) \frac{\partial^2}{\partial y^2} \ln p(x \mid y_t) \, dx \tag{185}$$

$$= 1 - \frac{1}{2\eta_t^2} \eta_t^2 \int p(x \mid y_t) \frac{\partial^2}{\partial y^2} (y_t - x)^2 \, dx + \eta_t^2 J(y_t) \tag{186}$$

$$= 1 - \int p(x \mid y_t) \frac{\partial}{\partial y} (y_t - x) \, dx + \eta_t^2 J(y_t) \tag{187}$$

$$= 1 - \int p(x \mid y_t) \, dx + \eta_t^2 J(y_t) \tag{188}$$

$$= \eta_t^2 J(y_t) \tag{189}$$

$$= \eta_t^2 \int p(x \mid y_t) \left( \frac{\partial}{\partial y} \ln p(x \mid y_t) \right)^2 \, dx \tag{190}$$

$$\geq 0 \tag{191}$$

where $J(y_t)$ is the Fisher information of $y_t$. Assume $\psi'(y_t) = 0$ for some $y_t$. Then

$$\frac{\partial}{\partial y} \ln p(X \mid y_t) = 0 \tag{192}$$

almost surely (Eq. 190). Then also

$$\frac{\partial}{\partial y} \ln p(X \mid y_t) = \frac{\partial}{\partial y} \ln \frac{p(y_t \mid x) p(X)}{\tilde{p}_t(y_t)} = \frac{\partial}{\partial y} \ln p(y_t \mid X) + \frac{\partial}{\partial y} \ln \tilde{p}_t(y_t) = 0 \tag{193}$$

or

$$\frac{1}{\eta_t^2} (y_t - X) = -\frac{\partial}{\partial y} \ln p(y_t \mid X) = \frac{\partial}{\partial y} \ln \tilde{p}_t(y_t) \tag{194}$$

almost surely. This implies $X$ is almost surely constant, that is, $p(x \mid y_t)$ is a degenerate distribution. But this contradicts our assumption that $p(x)$ is smooth. Since $p(y_t \mid x)$ is Gaussian with mean $x$ and therefore smooth as a function of $x$, $p(x \mid y_t) \propto p(x) p(y_t \mid x)$ must also be smooth. Hence, we must have $\psi'(y_t) > 0$ everywhere. Since $\psi(y_t)$ is strictly monotone it is also invertible.

Consider the squared Wasserstein metric,

$$W_2^2[P_X, P_{\hat{X}_{\text{MSE}}}] = \inf_{\hat{X}:\hat{X} \sim X} \mathbb{E}[(\hat{X} - \hat{X}_{\text{MSE}})^2] \tag{195}$$

where the infimum is over all random variables with the same marginal distribution as $X$ and which may depend on $\hat{X}_{\text{MSE}}$ (or equivalently may depend on $Y_t$). The solution to this problem is known from transportation theory to be $\hat{X}^* = \Phi_0^{-1}(\Phi_t(\hat{X}_{\text{MSE}}))$ (e.g., Kolouri et al., 2019), where $\Phi_0$ is the CDF of $X$, $\Phi_t$ is the CDF of $X_{\text{MSE}}$, and it is assumed that the measure of $X$ is absolutely continuous

with respect to the Lebesgue measure. We have

$$\Phi_0(x) = P(X \leq x) \tag{196}$$
$$= P(\tilde{F}_t^{-1}(Y_t) \leq x) \tag{197}$$
$$= P(Y_t \leq F_t(x)) \tag{198}$$
$$= P(\psi_t^{-1}(\hat{X}_{\text{MSE}}) \leq \tilde{F}_t(x)) \tag{199}$$
$$= P(\hat{X}_{\text{MSE}} \leq \hat{\psi}_t(\tilde{F}_t(x))) \tag{200}$$
$$= \Phi_t(\psi_t(\tilde{F}_t(x))), \tag{201}$$

implying

$$\hat{X}^* = \Phi_0^{-1}(\Phi_t(\hat{X}_{\text{MSE}})) \tag{202}$$
$$= \tilde{F}_t^{-1}(\psi_t^{-1}(\hat{X}_{\text{MSE}})) \tag{203}$$
$$= \tilde{F}_t^{-1}(Y_t) \tag{204}$$
$$= \hat{X}_F \tag{205}$$

and therefore that $\hat{X}_F$ is optimal.

Now let $M > 1$. Since $\mathbb{E}[\|\hat{\mathbf{X}}_F - \mathbf{X}\|^2]$ is invariant under the choice of $\mathbf{Q}$, we can assume $\mathbf{Q} = \mathbf{I}$ without changing the results of our analysis so that $\mathbf{X} = \mathbf{S}$ and $(X_i, Y_{ti}) \perp\!\!\!\perp (X_j, Y_{tj})$ for $i \neq j$. Since then

$$\ln p_t(\mathbf{z}_t) = \sum_i \ln p_{ti}(z_{ti}), \tag{206}$$

the ODE (Eq. 6) can be decomposed into $M$ separate problems

$$dz_{ti} = \left( -\frac{1}{2}\beta_t z_{ti} - \frac{1}{2}\beta_t \frac{\partial}{\partial z_{ti}} \ln p_{ti}(z_{ti}) \right) dt \tag{207}$$

for which we already know the solution is of the form $z_{ti} = (1 - \sigma_t^2)y_{ti}$ with

$$y_{ti} = \tilde{F}_{ti}(x_i) = \hat{\psi}_{ti}^{-1}(\Phi_{ti}^{-1}(\Phi_{0i}(x_i))), \tag{208}$$

where $\Phi_{ti}$ is the CDF of

$$\hat{X}_{i,\text{MSE}} = \mathbb{E}[X_i \mid Y_{ti}] = \mathbb{E}[X_i \mid \mathbf{Y}_t] = \hat{X}_{\text{MSE},i}. \tag{209}$$

On the other hand,

$$\inf_{\hat{\mathbf{X}}:\hat{\mathbf{X}} \sim \mathbf{X}} \mathbb{E}[\|\hat{\mathbf{X}} - \hat{\mathbf{X}}_{\text{MSE}}\|^2] = \inf_{\hat{\mathbf{X}}:\hat{\mathbf{X}} \sim \mathbf{X}} \sum_i \mathbb{E}[(\hat{X}_i - \hat{X}_{\text{MSE},i})^2] \tag{210}$$

$$\geq \sum_i \inf_{\hat{\mathbf{X}}:\hat{\mathbf{X}} \sim \mathbf{X}} \mathbb{E}[(\hat{X}_i - \hat{X}_{\text{MSE},i})^2] \tag{211}$$

$$\geq \sum_i \inf_{\hat{X}_i:\hat{X}_i \sim X_i} \mathbb{E}[(\hat{X}_i - \hat{X}_{\text{MSE},i})^2] \tag{212}$$

$$= \sum_i \inf_{\hat{X}_i:\hat{X}_i \sim X_i} \mathbb{E}[(\hat{X}_i - \hat{X}_{i,\text{MSE}})^2] \tag{213}$$

$$= \sum_i \mathbb{E}[(\Phi_{0i}^{-1}(\Phi_{ti}(\hat{X}_{i,\text{MSE}})) - \hat{X}_{i,\text{MSE}})^2] \tag{214}$$

$$= \sum_i \mathbb{E}[(\hat{X}_{F,i} - \hat{X}_{\text{MSE},i})^2] \tag{215}$$

$$= \mathbb{E}[\|\hat{\mathbf{X}}_F - \hat{\mathbf{X}}_{\text{MSE}}\|^2]. \tag{216}$$

That is, $\hat{\mathbf{X}}_F$ minimizes the squared error among all reconstructions achieving perfect realism. The second inequality follows due to the weaker constraint on the right-hand side; $\hat{\mathbf{X}} \sim \mathbf{X}$ implies $\hat{X}_i \sim X_i$ but not vice versa. Eqs. 214 and 215 follow from our proof of the case $M = 1$. $\qquad \square$

## H   COMPUTE RESOURCES

Training VDM took about 13 days using 32 TPUv3 cores (`https://cloud.google.com/tpu`).
No hyperparameter searches were performed to tune VDM for this paper. Training one HiFiC model
took about 4 days using 2 V100 GPUs and we trained 10 models targeting 5 different bitrates (with
and without pretrained weights). A few additional training runs were performed for HiFiC to tune the
architecture (reducing the stride) while targeting a single bitrate.

# I ADDITIONAL FIGURES

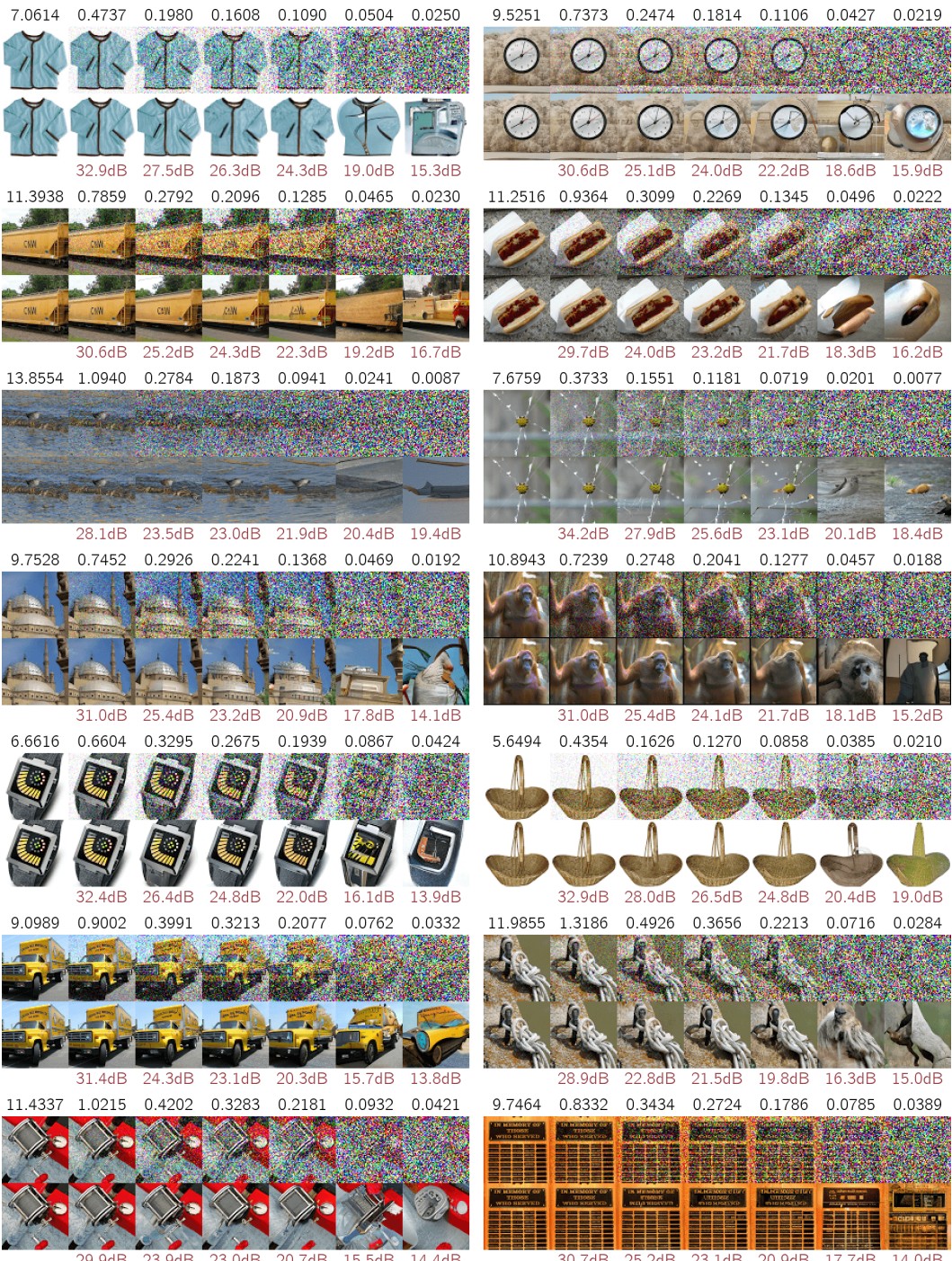

Figure 6: Top images visualize messages communicated at the estimated bitrate shown in black. The left-most bitrate corresponds to lossless compression with our VDM model. The bottom row shows reconstructions produced by DiffC-F and corresponding PSNR values in red.

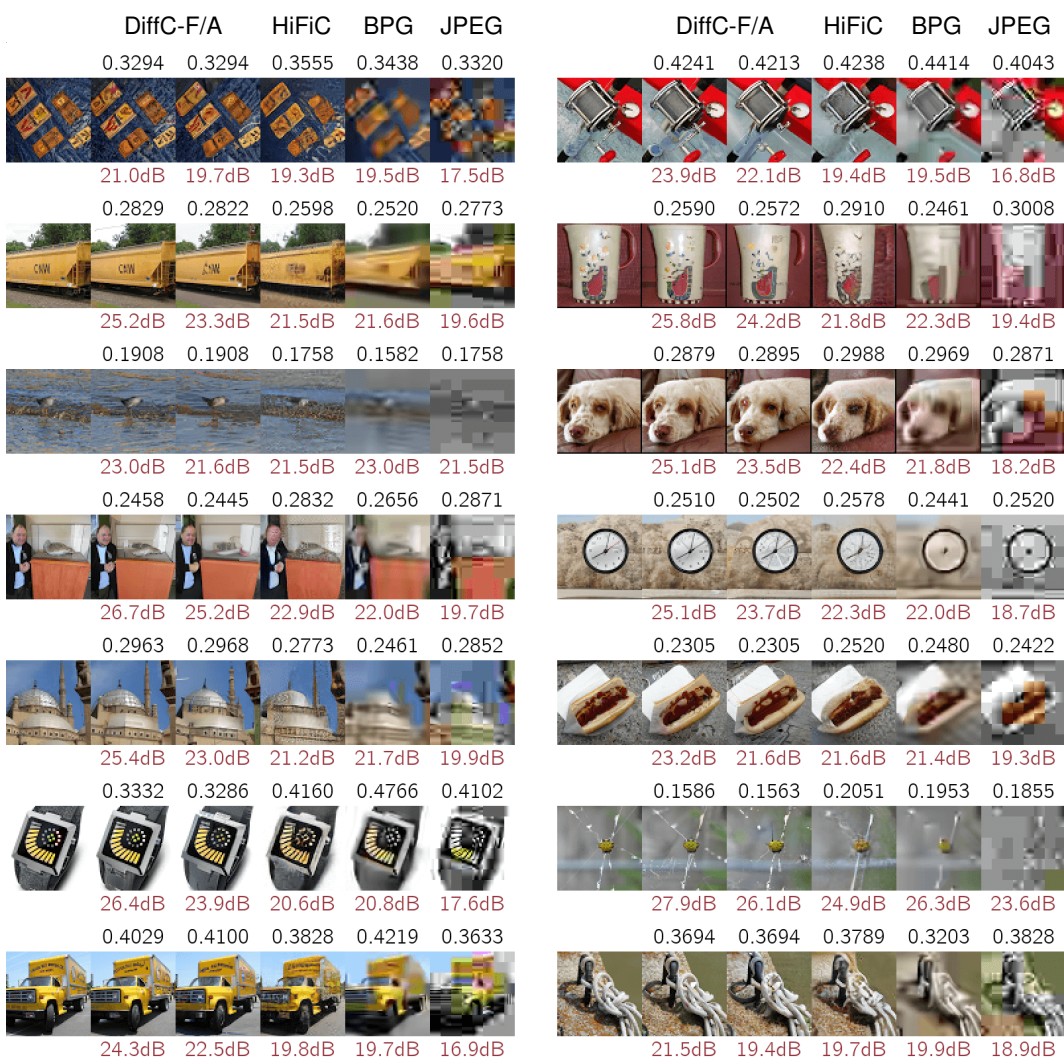

Figure 7: Additional reconstructions generated by different compression methods. The left-most column shows the uncompressed image. Bitrates are shown in black and PSNR values in red.

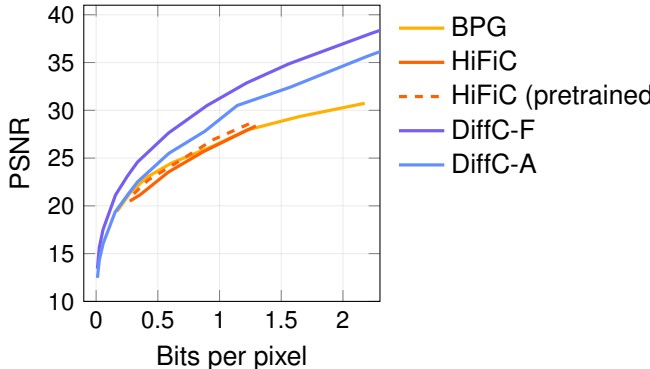

Figure 8: PSNR values in Section 5 were computed by calculating a PSNR score for each image and averaging. In contrast, this plot shows PSNR values corresponding to the average MSE.

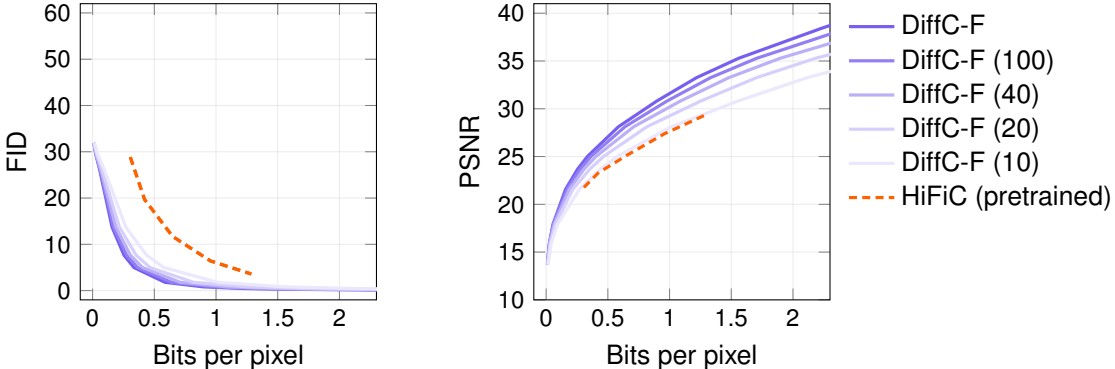

Figure 9: Performance relative to an upper bound on the coding cost when progressively communicating information in chunks of $B$ bits using the approach of Li and El Gamal (2018). The coding cost is estimated using $\frac{C_t}{B}(B + \log(B+1) + 5)$, where $C_t$ is the total amount of information sent (Eq. 5). At 10 bits the PSNR is comparable to our strongest baseline but the FID remains significantly lower.

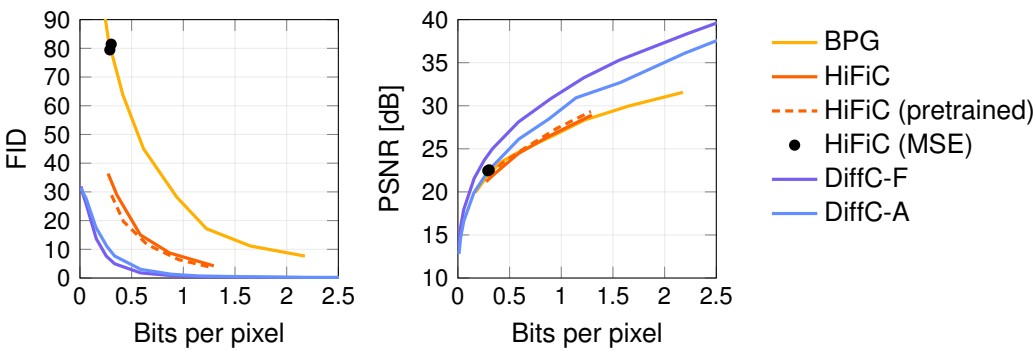

Figure 10: This figure contains additional results for HiFiC trained from scratch for MSE only. We only targeted a single bit-rate. The PSNR improves slightly while the FID score gets significantly worse.

