# OpenReview forum: "Lossy Compression with Gaussian Diffusion"
_ICLR.cc/2023/Conference — Submitted to ICLR 2023_

### Official Review · Reviewer_22JQ · 2022-10-23

**Confidence:** 4
**Correctness:** 4
**Technical Novelty And Significance:** 2
**Empirical Novelty And Significance:** 2
**Recommendation:** 5

**Clarity, Quality, Novelty And Reproducibility:**

### Clarity
The paper was very well written, and easy to read.

### Novelty
See third point under 'weaknesses' above.

### Reproducibility
The authors did not provide code (or promise to do so). It would therefore be difficult, but probably not impossible to reproduce the results from the paper.

**Strength And Weaknesses:**

### Strengths
 - The paper is very well written, and was enjoyable to read.
 - The paper was relatively honest about the limitations of the method.
 - I think the mathematical results were rigorous.
### Weaknesses
 - I'm concerned about the practicality of the methods presented in the paper, and particularly the use of methods like that of Li and El Gamal, which have computational complexity that scales exponentially with dimension (there are a number of other papers with which I have the same issue, such as Ho et al. 2020; Flamich et al. 2020; and Havasi et al. 2019). Normally we would refer to such methods as intractible, rather than merely 'computationally expensive' (pg. 4, fourth paragraph). The paper under review does explicitly mention the exponential scaling, but only in the 'Discussion' section at the end of the paper. Given that we are unlikely to get a perfect model for natural images (at least not this year), approximate realism is the best we will be able to achieve anyway, and therefore I want to know why we can't try a method which we can actually run, such as straightforward deterministic quantization of the latents, at least as a baseline.
 - Related to the above point, I was very disappointed that the authors didn't manage to implement their compression method, even for small-scale data. The comparisons to other methods, both quantitative and qualitative, seem unfair given that (as far as I know) all of those methods have actually been implemented, whereas the methods in the paper under review have not. Particularly the claims of 'simplicity' of the method. If it's really so simple then why not implement it?
 - I also felt that the novelty of diffusion model lossy compression with 'perfect realism' was limited. In particular I am comparing the paper to Ho et al. 2020, which presented rate distortion results (also not implemented, and IMHO problematic for similar reasons to the paper under review) for a diffusion model, the only difference being the 'perfect realism' constraint. IIUC Ho et al. mentioned that one could do this (though they don't use the term 'perfect realism') just below eqn. 15 in their paper. This is not a criticism of the mathematical analysis, which I did not have enough time to thoroughly comprehend, I hope other reviewers are able to comment on the novelty and significance of this.

### Minor points, typos and questions
 - In the first sentence of section 5, the authors say "we implemented DiffC based on VDM", perhaps this could be rephrased as "we measured the performance of DiffC based on VDM".
 - Typo in the second sentence of the second paragraph of section 5.2: "This is line" should be " This is in line".

**Summary Of The Paper:**

[NOTE: the citations I make below can all be found in the bibliography of the paper under review.]

This paper studies methods for doing lossy compression using a diffusion model, under the constraint of 'perfect realism'. 'Perfect realism' is defined in the introduction to the paper as the requirement that the marginal distribution of the reconstructions approximately matches the marginal distribution of the data (perhaps 'approximate realism' might be a better term for this property?).

A fairly detailed theoretical analysis was presented, as well as experiments which confirmed the analysis and showed that the method could perform well, assuming that the compression rate of a method in the paper of Li & El Gamal (2018) could actually be achieved. Unfortunately, the computational complexity of this method is exponential in the dimensionality of the latent space, and the authors did not attempt to implement actual compression.

**Summary Of The Review:**

This was a very well written paper and enjoyable to read, but I have serious concerns about its substance, particularly regarding the (im)practicality of the method presented, the fact that the authors were unable to implement the method, and the similarity to existing work, particularly Ho et al. 2020, therefore I judge it to be marginally below the acceptance threshold for ICLR.

---

### Official Review · Reviewer_sBei · 2022-10-25

**Confidence:** 4
**Correctness:** 3
**Technical Novelty And Significance:** 4
**Empirical Novelty And Significance:** 1
**Recommendation:** 5

**Clarity, Quality, Novelty And Reproducibility:**

The paper is well written and the theoretical analysis is thorough. The novelty is good. But, it may be difficult to reproduce the results, given that the implementation details are missing.

**Strength And Weaknesses:**

Strengths:

(1) The idea of using unconditional diffusion models for lossy image compression is novel. It has the striking feature that the encoding of an image can be done easily by corrupting the image with a (preferably non-isotropic) Gaussian noise.

(2) The theoretical analysis presented in this paper is thorough and rigorous.

(3) The authors made an early attempt to compare the proposed method with BPG and some learned codecs on ImageNet 64x64.

(4) The conclusion section of this paper is insightful, and provides a good summary of the pros and cons of the proposed method.

Weaknesses:

(1) The experimental results on ImageNet64x64 are inconclusive and less convincing. Apparently, BPG is not optimized for coding small-resolution images. This may also be true for HiFiC.

(2) While this is a good theoretical paper, the authors are alerted to a relevant work (Lossy Image Compression with Conditional Diffusion Models) that uses “conditional” diffusion models for lossy image compression. The proposed method is a direct contrast to this one.
https://arxiv.org/abs/2209.06950

(3) Complexity aspects are not fully addressed, although mentioned briefly.



**Summary Of The Paper:**

This paper proposes a new lossy image compression paradigm based on unconditional diffusion models. The noise corrupted image is compressed and sent to the decoder via reverse channel coding, and the reconstruction of the input image is done through the denoising process applied to the received noise corrupted image. Effort was made to understand the theoretical RD behavior of such a new coding paradigm. The authors also try to compare it with BPG and some other learned codecs on ImageNet 64x64.


**Summary Of The Review:**

This is a good theoretical paper. However, the experimental results on ImageNet64x64 are inconclusive and less convincing. BPG is not optimized for coding small-resolution images. This may also be true for HiFiC. As a result, it becomes unclear whether the proposed method is really worth further investigation. Complexity aspects are mentioned briefly but not addressed fully. It is expected that both the denoise process and reverse channel coding are complex. Their practicality is in question. Overall, this is an early attempt to use "unconditional" diffusion models for lossy image compression. It is to be noted that another late publication approaches lossy compression using "conditional" diffusion models and provides more empirical results.

---

### Official Review · Reviewer_JDSJ · 2022-10-25

**Confidence:** 3
**Correctness:** 3
**Technical Novelty And Significance:** 2
**Empirical Novelty And Significance:** 2
**Recommendation:** 3

**Clarity, Quality, Novelty And Reproducibility:**

This paper is written in a needless complex way. Unless further clarified, it is hard to justify the originality of this paper.

**Strength And Weaknesses:**

Strength:
1. The topic of using diffusion for lossy compression is interesting.

Weaknesses:
1. It’s hard to justify the originality of the proposed main methodology. The contribution is marginal. As far as I understand, all technologies of this paper is introduced from other paper, like diffusion and reserve channel coding.
2. Although the author claim they have some improvement, but they only compared with HIFIC.  As far as I know, there are plenty of lossy compression works every year [1, 2, 3]. The author should compare with these works, too.
3. The author claim in abstract, that this method is efficient. However, there’s no following experiment or number that can prove this claim.
4. The lossy compression techniques are also widely applied on video coding area, too. The author should add experiment on video compression.

[1] Cui Z, Wang J, Gao S, et al. Asymmetric gained deep image compression with continuous rate adaptation[C]//Proceedings of the IEEE/CVF Conference on Computer Vision and Pattern Recognition. 2021: 10532-10541.
[2] Xie Y, Cheng K L, Chen Q. Enhanced invertible encoding for learned image compression[C]//Proceedings of the 29th ACM International Conference on Multimedia. 2021: 162-170.
[3] Deng X, Yang W, Yang R, et al. Deep homography for efficient stereo image compression[C]//Proceedings of the IEEE/CVF Conference on Computer Vision and Pattern Recognition. 2021: 1492-1501.

**Summary Of The Paper:**

This paper proposed a diffusion based lossy image compression approach. In the evaluation, the proposed method outperforms a baseline.

**Summary Of The Review:**

As the major technical contribution of this paper is not clear, I recommend a reject. I would raise the score if the authors provide a good clarification.

---

### Decision · Program_Chairs · 2023-01-20

**Decision:**

Reject

**Justification For Why Not Higher Score:**

All the reviewers rated this paper below the acceptance threshold (scores were 3, 5, and 5). As the authors did not submit their response, their evaluations remained the same.

**Justification For Why Not Lower Score:**

N/A

**Metareview: Summary, Strengths And Weaknesses:**

This paper studies application of unconditional diffusion generative models to lossy compression under the constraint of "perfect perception", that is, reconstructions should have the same marginal distribution as the original data. Theoretical analysis under the assumption that the model has learned the data distribution perfectly is conducted in Section 4, results of which constitute the main strength of this paper. At the same time, reviewers were concerned with novelty of this paper, comparison with existing pieces of relevant work, and computational complexity of the proposal. These concerns have not resolved as the authors did not submit their responses.